# Imaging Constructs: The Rise of Iron Oxide Nanoparticles

**DOI:** 10.3390/molecules26113437

**Published:** 2021-06-05

**Authors:** Bianca Elena-Beatrice Crețu, Gianina Dodi, Amin Shavandi, Ioannis Gardikiotis, Ionela Lăcrămioara Șerban, Vera Balan

**Affiliations:** 1Advanced Centre for Research-Development in Experimental Medicine, Grigore T. Popa University of Medicine and Pharmacy of Iasi, 700115 Iasi, Romania; bianca.cretu@umfiasi.ro (B.E.-B.C.); dr.gardikiotis@yahoo.com (I.G.); 2BioMatter-Biomass Transformation Lab, École Polytechnique de Bruxelles, Université Libre de Bruxelles, 1050 Brussels, Belgium; amin.shavandi@ulb.be; 3Physiology Department, Grigore T. Popa University of Medicine and Pharmacy of Iasi, 700115 Iasi, Romania; ionela.serban@umfiasi.ro; 4Faculty of Medical Bioengineering, Grigore T. Popa University of Medicine and Pharmacy of Iasi, 700115 Iasi, Romania; balan.vera@umfiasi.ro

**Keywords:** iron oxide core, biopolymer shell, bio-inspired polymers, imaging techniques, in vivo, clinical trials

## Abstract

Over the last decade, an important challenge in nanomedicine imaging has been the work to design multifunctional agents that can be detected by single and/or multimodal techniques. Among the broad spectrum of nanoscale materials being investigated for imaging use, iron oxide nanoparticles have gained significant attention due to their intrinsic magnetic properties, low toxicity, large magnetic moments, superparamagnetic behaviour and large surface area—the latter being a particular advantage in its conjunction with specific moieties, dye molecules, and imaging probes. Tracers-based nanoparticles are promising candidates, since they combine synergistic advantages for non-invasive, highly sensitive, high-resolution, and quantitative imaging on different modalities. This study represents an overview of current advancements in magnetic materials with clinical potential that will hopefully provide an effective system for diagnosis in the near future. Further exploration is still needed to reveal their potential as promising candidates from simple functionalization of metal oxide nanomaterials up to medical imaging.

## 1. Introduction

Over the past century, the field of molecular imaging in living systems has expanded tremendously [1]. In general, molecular imaging modalities include, for example, magnetic resonance imaging (MRI), optical bioluminescence, optical fluorescence, targeted ultrasound (US), single-photon emission computed tomography (SPECT), and positron emission tomography (PET) [2]. Although all these molecular imaging modalities are available in clinics today, there is no single modality that is perfectly sufficient to obtain all necessary information for a particular question [3]. Hong et al. [1] established in their review that it is difficult to accurately quantify fluorescence signals in living subjects, particularly in deep tissues. MRI has high resolution, but it suffers from low sensitivity. Radionuclide-based imaging techniques have very high sensitivity but relatively poor resolution. Examples such as these could continue.

There are numerous review papers in the literature that describe imaging modalities and their available agents in detail. Therefore, we chose not to reiterate all these aspects in this paper. Instead, we focused on the essential feasibility and practicality of different imaging vehicles. Briefly: MRI is a non-invasive tool, and is generally used in clinics for high spatial resolution diagnostic imaging with the aid of conventional gadolinium compounds (such as T_1_-positive contrast agents and magnetic iron oxide nanoparticles as T_2_-specific agents that generate a bright or negative image where the compounds are accumulated. The main task of MRI contrast agents is to express short relaxation times—T_1_ and T_2_—which characterize the two independent processes of proton relaxation. These are longitudinal relaxation, which is responsible for bright images, and transverse relaxation, which is accountable for dark images [4]. The main drawbacks of gadolinium chelates are their non-specific distribution, fast elimination in tissue, accumulation in kidneys—which can impair function—and their limited ability to improve MRI imaging sensitivity. More details on MRI theory can be found in the review of Ellis et al. [5].

Functional imaging modalities such as SPECT and PET are situated at opposite corners of the electromagnetic spectrum, being based on γ-ray emissions. They provide 3D images of the administered radiopharmaceutical distribution and are dependent on the properties of radionuclides [6]. SPECT is more widely available than PET, but it is approximately ten times less sensitive. It is valuable because it enables concurrent imaging of multiple radionuclides. It is important to mention, as seen above, that SPECT radionuclides are simple to prepare and usually have a longer half-life than PET radionuclides; thus, they are available in a variety of chemical structures. Nonetheless, even if both techniques have quantitative advantages over MRI and optical imaging, the poor resolution of PET led to the design of hybrid imaging to track molecular events [7]. The most commonly used radionuclides in nuclear medicine include ^99m^Tc (metastable, short life of 6 h) and ^111^In (half-life of 2.8 days) for SPECT imaging, and ^64^Cu (t_1/2_ =12.7 h), ^18^F (t_1/2_ = 109.8 min) and ^68^Ga (t_1/2_ = 68.1 min) for PET [8].

To help physicians to look inside the human body and detect diseased tissue without the need for surgery, research in imaging is conducted in: (1) developing new tracers and (2) creating new technologies for accurate diagnosis. This review focused on the first objective, concerning the development of new tracers.

In this context, an important challenge in nanomedicine imaging is to design multifunctional agents that can be detected by single or multimodal techniques. An ideal nanoparticle imaging probe for next-generation, multifunctional tracers should have the following features: easy administration, excellent in vivo and radiolabelling stability [9], biocompatibility [10], selectivity, sensitivity [11], ability to observe accumulation in real-time and monitor disease progression [12], biodegradability or rapid excretion after imaging is complete, minimal-to-no side effects and cost-effectiveness—all while producing a strong imaging signal [13]. In recent years, various contrast/radiolabelled and/or fluorescent nanoparticles were developed as promising diagnostic and cancer evaluation tools [14]; however, multifunctional imaging agents simultaneously exhibiting all these features are extremely rare.

In the past 30 years, magnetic nanoparticles comprising an iron oxide core (usually magnetite- Fe_3_O_4_, maghemite- γ-Fe_2_O_3_ or hematite- α-Fe_2_O_3_) have attracted growing attention for their unique properties, e.g., magnetic functionality, surface-to-volume ratio, greater surface area, favourable toxicity profile and potential applications in biomedicine—especially as contrast agents. Iron oxide nanoparticles are at the forefront of science in the 21st century, boosted by the wide scope of their potential biomedical applications. This field is of tremendous importance, which can be confirmed by the scientific output of the last two decades and the more than 3000 scientific articles published each year, according to the Scopus platform.

The Molecular Imaging and Contrast Agent Database (MICAD) is an online source of molecular imaging and contrast agents that are under progress, in clinical trials or commercially available for medical applications. It was developed by the National Institutes of Health (NIH), sourced by the FDA (Food and Drug Administration) and published by the National Center for Biotechnology Information (US), (Bethesda, MD, USA) [15]. The database covers data published in peer-reviewed scientific journals from 2004 up to 2013, updated annually. It includes nearly 1444 agents developed for MRI, PET, SPECT, US, CT, optical, planar radiography, and planar gamma imaging. Briefly, this text presents—in MRI, SPECT and Multimodal sections—a chapter dedicated to iron oxide-based materials as contrast agents in various combinations. Table 1, Table 2 and Table 3 provide an overview of iron oxide-based materials that were developed as contrast agents during this period of time.

Taking into account all of the available presented data, the main objective of this review paper was to outline multifunctional magnetic tracers as single/multi-modality imaging probes. Specifically, we reviewed both the key technical principles of magnetic-based materials and the ongoing advancement toward an ideal contrast agent. First, this review examined iron oxide nanoparticle preparation approaches that are in current use in diagnostics as the core material. Second, this work reviewed multifunctional polymeric chain designs employed as the shell, followed by in vivo imaging evaluations on experimental animals using different imaging techniques and the current success of magnetic materials as contrast agents in clinical trials.

## 2. Techniques for Synthesis of Iron Oxide Core for Imaging Purposes

Iron oxide magnetic nanoparticle synthesis methods have enhanced the significant advances in magnetic materials applications, meaning that fabrication processes are significant pillars of the field. According to Ali et al. (2016) [16], more than 90% of magnetic nanoparticles are prepared using chemical methods. Nearly 8% are prepared using physical methods, and only 2% through biological processes. Each method has its own advantages and disadvantages; therefore, the physical and chemical properties of the obtained nanoparticles are dependent upon the conditions of fabrication.

Physical processes—such as aerosol/gas phase deposition, electron beam lithography, pulsed laser ablation, laser-induced pyrolysis and power ball milling—have the advantage of being easy to perform, but it is not possible to control the particles’ size in the nanometer region, resulting in irregular spheres. In the case of chemical techniques, the obtained nanoparticles may have irregular shapes, may be porous or nonporous, spheres, platelets, rod-like spheres, crystals, nanotubes, nanorods, bipyramids, facets, or other shapes. This is primarily dependent on the synthesis parameters, e.g., reagents, pH, temperature, and ionic strength. Chemical methods are simple, efficient, and manageable in size, shape and composition. Biological procedures, such as microbial incubation, are characterized by good reproducibility and scalability, and are low-cost, high yield for obtaining platelets and spheres. However, they are also laborious and time-consuming. Synthesis routes have been reviewed in many publications over the years in great detail; thus, just a few are briefly discussed here.

Many review papers have described different techniques for synthesis of iron oxide cores in general, or as comparative studies, pointing out the main properties, advantages and disadvantages for each method. As such, this review presents only magnetic particles reportedly in use as contrast agents [17,18,19,20,21,22].

A review worth mentioning is that of Niculescu et al. [23], which offered a general presentation of magnetite synthesis methods divided into conventional and unconventional procedures and correlated them with process outcomes, in terms of particle size, shape, magnetization properties and potential applications. Special attention was given to unconventional methods, i.e., microfluidic and recycled iron-based methods, from an alternative perspective. These methods are still under development but have already shown promising results in nanomedicine.

Another significant review paper is that of Caspani et al. [21], published in 2020. In addition to the principles associated with the use of contrast agents in MRI, their review discussed iron oxide nanoparticle synthesis methods, associated with their shape, properties, and their T_1_ and/or T_2_ capability.

Additionally, in the Fatima & Kim review, the summary of the frequently used preparation methods for iron-based nanoparticle synthesis displayed the advantages and disadvantages as they related to the shape and reaction temperature. They concluded that, although solvo & hydrothermal synthesis showed better control over shape and size, co-precipitation is still the predominately studied method [24].

Among these methods, the most frequently reported processes for the fabrication of iron oxide nanoparticles as diagnostic platforms were: co-precipitation, thermal decomposition, hydrothermal, sol-gel, and microemulsion, as presented in Figure 1. These were followed by polyol, magnetotactic bacteria, aerosol, microwave, electrochemical, and microfluidic methods, briefly described below.

### 2.1. Co-Precipitation

The co-precipitation method is the most implemented, simple, efficient, and (partially) eco-friendly chemical route used to synthesize iron oxides (either Fe_3_O_4_ or γ-Fe_2_O_3_) from aqueous Fe^2+^/Fe^3+^ salt solutions and a weak or strong base, at high temperature (from 20 to 150 °C) and under inert atmosphere. The chemical reaction for co-precipitation is depicted below:$${\mathrm{Fe}}^{2+}+2{\;\mathrm{Fe}}^{3+}+8{\;\mathrm{OH}}^{-}\;\to \;\mathrm{Fe}\;\left({\mathrm{OH}}_{2}\right)+2\;\mathrm{Fe}{\left(\mathrm{OH}\right)}_{3\;}\to {\mathrm{Fe}}_{3}O_{4}+4{\;H}_{2}O$$

Khalafalla et al. [25] proposed a co-precipitation method for the synthesis of aqueous magnetic fluid for the first time in 1980, followed by Massart [26] in 1981, who reported roughly spherical magnetite (Fe_3_O_4_) particles with a diameter of 8 nm as measured by X-ray diffraction analysis. Further studies, developed in this area over the years, indicated that the size, shape and iron oxide nanoparticle composition depended on several parameters, including: the Fe^2+^/Fe^3+^ ratio, the type of salts used in the reaction (e.g., chlorides, sulphates, nitrates, perchlorates), temperature reaction, pH value, ionic strength, base type (NaOH, Na_2_CO_3_, NH_4_OH), stirring rate, inlet of nitrogen/argon gas, and flow rate of the basic solution [27]. The most important steps to achieve a complete precipitation are: to maintain the pH range between 8 and 14, a stoichiometric ratio of Fe^3+^/Fe^2+^ at 2:1 and a non-oxidizing (oxygen-free) environment [28]. However, the obtained bare nanoparticles usually have the tendency to aggregate, mainly due to high surface-area-to-volume ratio, strong magnetic attraction among particles, van der Waals forces and high surface energy [29]. The addition of anionic surfactants as dispersing agents and surface coatings with polymers, proteins, starches, non-ionic detergents or polyelectrolytes can control the size of the nanoparticles and stabilize them. If all these parameters are properly adjusted, it is possible to tailor the characteristics of iron oxide nanoparticles and to obtain uniform magnetic particles with a size ranging from 17 to 2 nm, magnetic functionality, high surface area and a non-toxic profile.

To obtain magnetic colloidal ferrofluids with suitable properties for imaging aims, bare magnetic nanoparticles are coated with different biocompatible agents, represented by inorganic materials (e.g., silica, gold, or gadolinium), or polymer stabilizers (e.g., dextran, carboxydextran, carboxymethylated dextran, PEG, PVA, chitosan), which provide high colloidal stability against aggregation in biological media and/or allow tailoring the surface properties and coating. This is discussed further in the next section.

### 2.2. Thermal Decomposition

This method involves the decomposition of organometallic precursors, such as metal acetylacetonates [M(acac)_n_], (M = Fe, Mn, Co, Ni, Cr; n = 2 or 3, acac = acetylacetonate), metal cupferronates [M(Cup)_x_] (Cup = *N*-nitrosophenyl hydroxylamine) or metal carbonyls, in high-boiling organic solvents (diphenyl ether, diethylene glycol diethyl ether, dimethyl formamide, benzyl ether, octadecene, chloroform), in the presence of stabilizing surfactants such as fatty acids, oleic acid and hexadecylamine [30]. The thermal decomposition method implies high temperatures, around 280–350 °C, and an inert atmosphere (nitrogen/argon). The parameters that can tune the size, morphology, and chemical stability of the produced nanoparticles are: the type and ratio of the precursors, surfactants, and solvents, temperature, time, and aging phase [31]. Magnetic iron oxide particles synthesized by this technique are characterized by a better control over size and shape, high level of monodispersity and good crystallinity compared to the co-precipitation method. The synthesis process time is approximately an hour and magnetic nanoparticles are collected by centrifugation, but the high temperatures, expensive toxic reagents, and the laborious purification steps of the final product all hamper its implementation in biomedical applications.

### 2.3. Hydrothermal

The hydrothermal method yields magnetic nanoparticles with very good size and shape control and relatively broad size distribution from aqueous solutions of metal salt precursors (ferric nitrate, ferric chloride, ferrous oxalate, potassium ferrocyanide) dissolved along with surfactants/capping agents (sodium dodecylsulfonate (SDS), sodium dodecylbenzene sulphonate (DBS), cetyltrimethyl ammonium bromide (CTAB), hexadecylpyridinium chloride (HPC), polyvinylpyrrolidone (PVP)) in a reactor or autoclave at high temperatures (from 130 °C to 250 °C) and high pressures (from 0.3 to 4 MPa) under inert atmosphere (nitrogen/argon flow) [32]. The reaction parameters [33]—e.g., temperature, pressure, pH, solution concentration, precursor type and concentration, and residence time—define the morphology and particle size, as well as the possibility of growing crystals of varied shapes [30]. This adaptable method has the advantage of simplicity without requiring special reagents, but its need for high pressures and temperatures reduces the scalability and potential biomedical applications.

### 2.4. Sol-Gel

The sol-gel technique is a relatively simple and cost-effective route to synthesize nanostructured monodispersed metal oxides that are precisely controlled in size, shape and internal structure, with a pure amorphous phase and homogeneity. This method is based on the formation of colloidal sol using hydrolysis and condensation of the precursors, metal alkoxides (ferric nitrate, ferric chloride, ferrous chloride, ferrous sulphate), in ethanol/water solution. Using additional condensation and inorganic polymerization, the sol is then gelled in order to obtain three-dimensional metal oxide networks [34]. As these reactions are conducted at ambient conditions, the formed gels need extra heat treatments to obtain the final crystalline state. The parameters that influence the properties and the structure of the iron oxide nanoparticles are: the precursor concentration, solvent nature, temperature, pH, kinetics, properties of the gel and mechanical stirring rate. The associated disadvantages include high permeability, weak bonding and low wear resistance [17].

### 2.5. Microemulsion

This method involves dispersion of two immiscible liquids (oil-in-water (o/w) or water-in-oil (w/o)) separated by an interfacial film of surfactant molecules. The reaction mixture is composed of an aqueous Fe^2+^/Fe^3+^ solution phase subjected to sonication and heat at a particular temperature (in the range of 20 to 80 °C) and the other, containing organic solvents (propyl alcohol, heptane, cyclohexane) and stabilizing surfactants (e.g., Tween 20 and Tween 80, DSS, SDS, AOT and CTAB) [27]. The surfactant is an amphiphilic molecule that lowers the interfacial tension between oil and water phases. Magnetic iron oxide particles synthesized through this method are characterized by good control over the particle shape, relatively narrow size distribution and crystalline structure. The type and structure of the surfactant, along with physiological conditions (near ambient conditions of temperature and pressure) modulate the magnetic nanoparticles. The main advantage of this two-phase-method is the narrow particle size distribution, but the low yields of nanoparticles, the large amount of solvent required for their synthesis and the presence of residual surfactants create barriers in scale-up procedures and specific functions [35].

### 2.6. Other Used Methods

Polyol synthesis turned out to be a versatile wet-chemistry method for the synthesis of magnetic nanoparticles with variable shapes, sizes and compositions, using a liquid organic compound (polyols, such as ethylene/diethylene/tetraethylene glycol, 1,2 propylene glycol, PVA or 2-pyrrolidone) acting both as a solvent and a reducing agent. The basic principle of the polyol method was explained by Caruntu et al. [36], as a two-step procedure; first, the formation of the hydroxides, followed by metal centres chelation. Through simple optimization of the operating synthesis conditions—namely, the nature of the magnetic precursors and precipitator, the solvents’ nature, the addition of an extra stabilizer, and the temperature, pressure and duration of the reaction—the metal nanoparticles features can be tailored for the targeted application [37,38]. Compared with thermal decomposition or co-precipitation methods, the polyol process yields nanoparticles with a narrow particle size distribution in a simple, environmentally friendly, reproducible and cost-effective way, without the need for an inert atmosphere [39]. This method, considered an iteration of the solvothermal approach, is not without drawbacks; the formed particles lack homogeneity [20].

Another eco-friendly method used to produce magnetic nanoparticles for use as imaging contrast agents is magnetotactic bacteria, a group of Gram-negative bacteria called magnetosomes [40]. Magnetotactic bacteria are defined as intracellular nanocrystals of the magnetic minerals (magnetite or greigite) surrounded by a phospholipid bilayer membrane [41]. During biosynthesis, the magnetosomes are aligned in a linear sequence by connecting them to a cytoskeletal filament, which permits the bacteria to navigate along the geomagnetic field. The magnetosomes’ features are controlled by a specific set of parameters, e.g., genes (different species) that encode proteins and maintain the structural integrity or transport iron, magnetotactic bacteria isolation (sonication, treatment with sodium hydroxide, press, pressure homogenizer), purification (magnetic separation, proteinase K treatment), sterilization, heating, and so on. Compared with chemically synthesized magnetite nanoparticles, magnetosomes are of high chemical purity, display a narrow size range, are highly uniform and are permanently magnetic at ambient temperatures as stable single-magnetic domain crystals. Because of these physical/chemical properties, magnetosomes are used in many biomedical applications, but the main concerns of researchers are linked with the safety index—since the introduction of magnetosomes or the magnetosome reporter gene into cells can have adverse effects on cells over the 200 µmol/kg dose [42].

Since the first synthesis of γ-Fe_2_O_3_ by aerosol spray pyrolysis in 1993, the method grew into a promising, cost-effective, and scalable procedure that led to high particle production levels [43]. This approach involves the spraying of precursor salts into a hot reactor, where they are condensed as small droplets in vapor form, a parameter that determines the size. Other important factors that control the size of the particles are precursor composition, solvent nature, rate of evaporation, time spent in the reactor, and temperature. The technique produces spherical nanoparticles with narrow size distribution, high homogeneity and monodispersity, and it has a high yield. However, it falls short due toto the particles’ inner structure, pore size uniformity and extremely high temperatures [44].

Electrochemical synthesis of magnetic iron oxide nanoparticles like maghemite and magnetite involves passing electric current between electrodes (anode and cathode) located in an electrolyte. The anode is oxidized to metal ion species, reduced to metal by the cathode with the assistance of stabilizers, and then deposited in the form of a coating or thin film on the electrode [19]. The parameters that are used to obtain unique products (which are not possible to obtain using other methods) relate to the current passed through the cell, cell potential, oxidizing or/and reducing power, bath composition, pH, electrolysis type, electrolyte concentration and composition. This method does not require high temperatures, and has the advantage of control over particle size, but it is complicated and lacks reproducibility. Additionally, it is prone to the presence of amorphous impurities due to poorly ordered products [45].

The microfluidic method first appeared as a proposed solution to the identified drawbacks of existing synthesis procedures, and as an alternative to traditional reactors [46]. The microfluidic synthesis strategy is based on the fluid movement within micro-scaled channels with unique geometries that allows for improved control over reaction conditions and real-time, in-line characterization [47]. The microfluidic systems used for the synthesis of magnetic cores are single-phase flow reactors, due to their homogeneity and versatility in controlling process parameters and droplet-based microreactors as well as their rapid production and analysis of reproducible and scalable particles with specific sizes, shapes, and morphologies. The obtained nanoparticles are known for their narrow size distribution, uniform shape, improved reproducibility, shorter reaction time, increased yield-formation of pure phase magnetite, and friendly reaction conditions with no extra additives or heating. However, since the field of microfluidics as applied to nanomedicine is still in its infancy, there are several drawbacks associated with this method, namely: surface roughness, limited production rate, possibility of clogging micro-channels, leaks—leading to experimental failure, capillary force and chemical interactions [48,49].

## 3. Surface Shell Engineering of Magnetic Materials for Imaging Purposes

Many approaches have focused on the encapsulation of iron oxide nanoparticles with biocompatible materials forming a composite morphology often referred to as a core-shell structure [50]. Bare surface iron oxides agglomerate due to strong magnetic attraction between particles, van der Waals forces and high surface energy, as mentioned above. Additionally, they endure rapid elimination by the RES when exposed to biological media or can be toxic in high concentrations due to iron dissolution [51]. All these drawbacks are limited when magnetite nanoparticles are embedded in a non-magnetic matrix. The appropriate coatings can stabilize them in a physiological environment, provide chemical functionality for extra modifications and control the particle size and shape. Furthermore, the shell addition and geometric arrangement improve the biocompatibility of the material as well as the biokinetics and biodistribution in the body. The nature of the surface coating of magnetic nanoparticles depends on the application and required functionalities. Polysaccharides, synthetic polymers, proteins, enzymes or antibodies bind to hydrophobic surfaces with a large surface-area-to-volume ratio of the iron oxide cores in order to design core-shell materials with additional chemical and physical functions for specific applications [52].

Several review reports can be found in the literature that deal with the most important topic in the design of iron oxide nanoparticles as diagnostic agents: functionalization/shell coating/surface modification. This topic and its associated methods offer high stability in physiological media, stealth, biological ligand-binding ability, biocompatibility, non-toxicity, enhanced physiochemical and mechanical properties and improved dispersion [53,54,55,56].

In 2008, Laurent et al. [17] summarized different stabilization methods of magnetic iron oxide nanoparticles using monomeric stabilizers (carboxylates and phosphates), inorganic materials (silica and gold) and polymeric stabilizers (dextran, PEG, PVA, alginate, chitosan, PEI, PEI-PEO-block PGA, PLGA, PVP, PAA, poly(ε-caprolactone), sulphonated styrene-divinylbenzene, polymethacrylic acid, PEO-*b*-poly(methacrylic acid), polyalkylcyanoacrylate, or arabinogalactan) as their main strategies to obtain magnetic colloidal ferrofluids that were stable against aggregation in both biological media and magnetic fields. Another strategy mentioned by the authors to synthesize polymeric core/shell magnetic nanoparticles is to use preformed synthetic polymers as a matrix to control the formation of magnetic cores, e.g., ABC triblock polymer (polyisopropene-block-poly (2-cinnamoylethyl methacrylate)-block-poly(*tert*-butyl acrylate)), polystyrene-polyacrylate copolymer gel template, poly(methylmethacrylate)/polypyrrole bilayers and in situ polymerization of iron oxide nanoparticles in poly(styrene/acetoacetoxyethylmethacrylate) particles.

In 2016, Narayanaswamy et al. [57] published a review which described, in a chapter focused on nanoparticle-based imaging agents, different nanobiomaterials for image enhancement. Their review covered developments up to 2010, based on the coating type but not on the magnetic core. After a brief representation of these, the authors presented polysaccharides and proteins—such as albumin, alginate, apoferritin, beta-glucan, casein, cellulose, chitosan, chondroitin-sulfate, collagen, cyclodextrin, dextran, fibrinogen, fucoidan, gelatin, heparin, hyaluronic acid, lectin, mannan, mannose, pullulan, starch and zein-based nanoparticles—widely used as imaging agents.

More recently, in 2020, the review paper of Avasthi et al. [20] addressed key aspects in the development of IONPs only for MRI applications—namely, the synthesis of the magnetic core, functionalization processes and in vivo studies with emphasis on tumour models. Their chapter on the functionalization of IONPs was divided according to the methods used to functionalize IONPs for clinical purposes, e.g., in organic supra-structures, inorganic coverage and ligand exchange. The first class of highly branched macromolecules were defined by 3D architectures, and included nanomicelles, dendrimers, liposomes and nanogels, developed to create hybrid materials for imaging and therapy. In their review, the authors described the use of polymeric folate-conjugated *N*-palmitoyl chitosan and PEG-phosphatidylethanolamine micelles, poly (amidoamine) dendrimer, 1,2-distearoyl-snglycero-3-phosphoethanolamine phospholipids liposomes, copolymer [*N*-isopropylacrylamide, methacrylic acid and poly (ethylene glycol) methacrylate] and alginate-PEI nanogels for the encapsulation/conjugation of iron oxide nanoparticles. The inorganic materials for IONPs core coating described in this review paper were: mesoporous silica and its complexes with *N*-isopropylacrylamide, *N*-(hydroxymethyl) acrylamide, *N,N*′-methylenebis(acrylamide) monomers, PVP and PEG molecules. The ligand exchange method, as expressed by the authors, required multiple interactions, potentials/forces, and the use of reactive binding molecules, making this coating strategy very complicated.

A list of commonly used SPION coating materials (as well as their sizes and properties) was presented by Nelson et al. [58] in their educational review paper, highlighting the associated challenges in the use of SPIONs in multiple imaging modalities. This review represented an important guide for young scientists, illustrating the basic concepts of MRI, the basic construct of SPIONs, in vitro challenges, shapes and sizes, various coatings with different materials, and application of SPIONs in diagnostics and therapy.

As presented above and in Figure 2, of the multitude of polymeric shells used to adjust the size and surface nature of iron oxide core nanoparticles—which are essential in the development of a successful diagnostic platform—only a few are summarized in Tables 4 and 5 and succinctly detailed in the next sections.

### 3.1. Biopolymeric Shells

Various natural polymers such as dextran [59,60], chitosan [61], hyaluronic acid [62], starch [63], albumin [64], alginate [65], gelatin [66] and polydopamine [67,68] (see Table 4) have been used as coating materials for superparamagnetic iron oxide nanoparticles, also known in the literature as SPIONs, to reduce aggregation, enhance their stability and biocompatibility [27], features also mentioned above.

#### 3.1.1. Chitosan and Its Derivatives

Among polysaccharides, chitosan stands out as a biocompatible polymer, and one widely explored in biomedical applications due to its plentiful advantages. These include nontoxicity, controlled biodegradability, and antioxidant and antimicrobial activity [152,153]. Its functionality and ability to form nanoparticles and nanocapsules with iron oxides or other inorganics has extended its use for diagnostic, hyperthermic, cancer therapeutic or theranostic purposes [154,155,156]. Great interest has been devoted to the development of magnetic chitosan nanostructures with such remarkable features as MR imaging aids, especially as T_2_ contrast agents.

Studies on chitosan-iron oxide cores as nanobiomaterials for image enhancement from the 2000s were detailed in a chapter by Narayanaswamy et al. [57]. Therefore, we presented significant reports from 2010 up to the present below.

Several chitosan derivatives have been proposed as coating materials for SPIONs endowed with better contrast ability in MR imaging. For instance, glycol chitosan, a fully soluble derivate in neutral and acidic media, was employed to increase the steric stabilization and aqueous solubility [157]. In this regard, Lee et al. [147] prepared biodegradable magnetic nanoparticles composed of a hydrophobic core (PLGA and SPIO) and a hydrophilic shell (glycol chitosan) utilizing an emulsion-diffusion-evaporation technique. The nanoparticles were internalized in cells and accumulated in lysosomes. A high level of radioactivity was observed in the liver, shortly after intravenous administration of the ^99m^Tc-labeled magnetic nanoparticles. Through in vitro and in vivo tests, the authors proved that magnetic nanoparticles could be useful as an efficient contrast agent for MRI, as they were able to be degraded after serving their imaging function.

Likewise, Xiao et al. [121] developed a tumour-targeted MRI nanosystem composed of iron oxide nanoparticles encapsulated in self-assembled micelles, based on folate-conjugated *N*-palmitoyl chitosan. They showed their tumour-targeting ability through in vitro and in vivo tests. Their results indicated that the signal intensities of T_2_-weighted images in established HeLa-derived tumours were significantly reduced, indicating that folate-functionalized micelles could serve as safe and effective MRI contrast agents for detecting folate receptor-positive tumours. Hobson et al. [144] prepared clustering SPIONs by encapsulation of hydrophobic iron oxide nanoparticles in an amphiphilic chitosan derivate, namely *N*-palmitoyl-*N*-monomethyl-*N*, *N*-dimethyl-*N,N,N*-trimethyl-6-*O*-glycolchitosan. Clustered SPIONS exhibited a high spin-spin (r_2_) to spin-lattice (r_1_) relaxation ratio (r_2_/r_1_) that induced a superior contrast ability, accumulated only in the liver and spleen after intravenous administration and provided clear MRI images of the liver vascularization (when compared with Ferucarbotran, a commercially available contrast agent).

Biocompatible SPIONs coated with a layer of cationic derivative of chitosan and hyaluronic acid−curcumin conjugate with very high values of saturation magnetization (43.4 ± 0.2 A·m^2^/kg Fe) and transverse relaxivity (469.7 ± 2.3 mM^−1^·s^−1^) were developed by Lachowicz et al. [148] The coated SPIONs could be considered as bimodal agents for both MRI and fluorescence detection. In a further study, comprehensive results on uptake and bioreactivity of charged chitosan-coated SPIONs with high stability, designed as T_2_ contrast agents, were reported by Kania et al. [120] in 2018. The authors explored the in vivo bioreactivity of SPIONs coated with either cationic (low molecular weight chitosan with quaternary ammonium groups, DS-57%) or anionic chitosan derivatives (carboxymethyl chitosan substituted with sulfonate groups, degree of substitution of 66%) using a BALB/c mouse model. The overall results proved that the kidneys and liver were the organs involved in SPIONs removal. It is noteworthy to mention that chitosan-coated SPIONs could be employed for long-term studies as they showed liver-enhancing MRI contrast even 7 days after administration.

Additionally, amphiphile chitosan derivatives were developed as suitable coating materials for SPIONs, planned for imaging purposes. For instance, oleyl chitosan-coated iron oxide nanoparticles were recommended as a dual probe for optical and magnetic resonance imaging of tumours [107]. In a recent report, Hemalatha et al. [158] showed that oleyl chitosan was a suitable platform for inorganic nanoparticle encapsulation (namely hybrid iron oxide/gold nanostructures) that guaranteed both high colloidal stability and better magnetic resonance imaging. At the same time, the magnetic nanocomposite exhibited good appropriate characteristics for computer tomography: biocompatibility, X-ray attenuation properties and hemocompatibility.

On the other hand, Khmara et al. [72] formulated chitosan-stabilized iron oxide nanoparticles in two steps: (i) co-precipitation method and (ii) consequent polymer coating, in the presence of urea (used to support a uniform distribution of iron oxide nanoparticles’ size). The MRI tests showed a significant prevailing effect on T_2_ (the transversal relaxation time) with high transversal relaxivity values (r_2_ = 238.16 mM^−1^·s^−1^) that exceeded those of clinically-used iron-based contrast agents (namely Feridex r_2_ = 120 mM^−1^·s^−1^, Resovist r_2_ = 186 mM^−1^·s^−1^ and Combidex r_2_ = 65 mM^−1^·s^−1^ [159]. In a recent report, Sun et al. [160] investigated the in vitro ability of chitosan iron oxide nanoparticles to be used for dual-mode US/MR imaging and concluded that they showed potential as efficient contrast agents.

In order to provide a more detailed in vivo diagnosis, Chung et al. [161] prepared an MRI and NIRF multimodal imaging system based on glycol chitosan-coated SPIONs and functionalized it with matrix metalloproteinase sensitive peptide conjugated with black hole quencher 3 and Cy5.5 dye at each end. NIRF-based optical techniques provided data regarding the biological events that occur at a molecular level through the degradation of activated linkers or molecular triggers, e.g., matrix metalloproteinase (MMP)-sensitive peptide. Magnetic nanosystems showed a maximum NIRF intensity at 48 h post-injection in tumour tissue (which was approximately 8 times higher than other organs) and a relative MRI contrast enhancement compared to normal muscles. Concurrently, magnetic nanosystems depicted the anatomic image of the tumour site and provided MMP-2-dependent biological data.

#### 3.1.2. Dextran and Derivatives

Dextran is a polysaccharide consisting of linear chains of α-1,6 linked glucopyranose residues with α-1,3 or 1,4 side chains linked to the backbone units, in varying proportions and sequential arrangements with excellent water solubility, biocompatibility, biodegradability, improved transfection efficiency, low-cost and non-toxicity, suitable for use in biological systems.

Dextran has been used as a polymer coating material for iron oxide nanoparticles since the early 1980s. The first report of the formation of magnetite in the presence of dextran was by Molday and Mackenzie [162]. Five years later, Magin et al. [163] studied dextran magnetite as a liver tumour contrast agent for MRI, with both T_1_ and T_2_-weighted spin-echo images after intravenous injection on Fischer 344 rats. In 1996, the FDA approved Feridex I.V. (ferumoxides) Advanced Magnetics company as the first nanoparticle-based iron oxide imaging agent to detect liver lesions. Feridex™I.V.^®^ (ferumoxides injectable solution) is a sterile aqueous colloid of superparamagnetic iron oxide associated with dextran for intravenous administration as an MRI contrast media.

Dextran and its derivatives (e.g., carboxymethyl dextran, carboxydextran, aminodextran and dextran sulphate) were layered in iron oxide nanoparticle cores as early as 2000 and up to 2014, and were used as versatile platforms for MRI for tumour imaging (especially pancreatic and colon cancer cells), for in vitro labelling of cells and subsequent cell-tracking of phagocytic cells in vivo in liver fibrosis, atherosclerosis, lung cancer, head and neck cancer cells, etc., as described in Narayanaswamy et al. [57] and Table 4.

In the same period, a few of the SPIONs coated with dextran (e.g., Endorem, Resovist or Clariscan) were approved as clinical liver/lymph node MRI-based contrast agents. However, in 2009–2011, these products were withdrawn from the market due to insufficient clinical trial results and major safety concerns. These issues are discussed in the next section.

A contrast agent for both CT and MRI was proposed by Naha et al. [101] that synthesized dextran-coated bismuth-iron oxide nanoparticles and was used for in vivo imaging experiments in wild-type C57BL/6J inbred mice using a micro-CT and a 9.4 T MRI system. As shown in the experimental article, the composite nanoparticles were present in the heart, blood vessel and bladder, and were excreted via urination, with significant concentrations in the kidneys and urine. Substantial signal loss in T_2_-weighted MR images was also observed in the liver at 2 h post injection. Considering both positive in vivo CT and MR imaging results, the dextran-coated bismuth-iron oxide nanoparticles could be used as a dual modality contrast agent.

Valuable results were obtained by Wabler et al. [164] that established a relationship between MRI signal intensity and iron content for formulations, e.g., human prostate carcinoma DU-145 cells loaded with starch-coated, bionized nanoferrite, iron oxide (Nanomag^®^ D-SPIO), Feridex™ and dextran-coated Johns Hopkins University particles (NanoMaterials Technology, Singapore). The MRI data showed a linear correlation between increased iron content—quantified using inductively-coupled plasma mass spectrometry—and decreased T_2_ times.

The surface charge in the range of surface (−1.5 mV to +18.2 mV) of dextran-based SPIONs was tailored for increased uptake and MRI contrast of mesenchymal stem/stromal cells by Barrow et al. [126] using fluorescein isothiocyanate and diethylamino ethyl compounds. The in vitro MRI tests acknowledged that this functionalization strategy controlled the safe uptake into stem cells, which was a required condition before clinical evaluation.

Two types of materials—based on multicore superparamagnetic iron oxide nanoparticles stabilized with dextran and PEG-gallic acid polymer—were synthesized by Ziemian et al. [130] in 2015, in order to develop an optimized tracer for MPI, a new imaging modality, initiated by Gleich and Weizenecker [165] in 2005. The authors compared the capabilities of the two materials using Resovist agent, and demonstrated excellent MPI potential, superior to what was commercially available.

Bombesin peptide analogue (KGGCDFQWAV-βAla-HF-NIe), covalently attached to the dextran SPIONs, were used for the first time in 2015 as a new targeting MRI contrast agent for breast cancer detection [77]. The MRI study (on a 1.5 T MRI Scanner) indicated that this new contrast agent showed T_2_* values at 13 and 30 h after the tail vein injection and demonstrated the ability to accumulate within a breast tumor.

The detection of lung cancer metastasis by MRI was enhanced using SPIONs coated with oleic acid and carboxymethyl dextran, conjugated to mouse anti-CD44v6 monoclonal antibody. After the determination of their physicochemical properties, the in vitro MRI studies detected human lung adenocarcinoma (A549) cells in T_2_ relaxation time [135].

Commercially available dextran-coated SPIONs of nearly 215 nm and positive surface charge (FeraTrack Direct) were evaluated for direct labelling of stem cells and in vivo MRI tracking [166]. The nanoparticles were labelled with bone marrow-derived mesenchymal stromal cells and neural stem cells and injected to C6 glioma-bearing nude mice. The tracers were detected as hypointense regions within the tumour using 3 T clinical MRI up to 10 days post injection. These results were also confirmed by histological analysis.

Carboxymethyl-diethylaminoethyl dextran magnetite particles were synthesized as a blood-pooling, non-gadolinium-based contrast agent, since it is well known that gadolinium-based agents increase the risk for nephrogenic systemic fibrosis in severe renal insufficiency patients. Both positive (+9.6 mV) and negative (−10.4, −41.0 mV) surface charge particles were injected into Japanese white rabbits to evaluate if the degree of charge altered the blood-pooling time. The in vivo pooling time was prolonged for up to 300 min for all three differently charged particles, thus exhibiting prolonged vascular enhancing effects [167].

The following year, a positively charged dextran-coated SPION containing diethylaminoethyl and fluorescein isothiocyanate was used to efficiently label macrophages for MRI–based cell tracking in vivo up to 3 weeks post-transplantation. The labelling was more efficient than ferumoxytol, but the authors considered that further studies were required to determine if SPION-labelled bone marrow–derived macrophages had a therapeutic effect in liver disease [168].

Five different sizes, ranging from 30 to 130 nm of dextran-coated SPIONs were fabricated by cold gelation method [129] and tested for their imaging properties by T_2_, T_2_* and T_1_ relaxation times with a 7 T MRI in vitro in agarose gels. Independent of their size, the nanoparticles displayed their safety and internalization by macrophages and the absence of hypersensitivity reactions—both properties suitable for future clinical development as MRI contrast agents [60].

A facile method to develop iron oxide nanoparticle-loaded magnetic dextran nanogels as an MRI guided nanoplatform was proposed by Su et al. in 2019 [74]. First, the iron oxide nanoparticles were pre-synthesized by co-precipitation method, followed by physical blending with aldehyde dextran solution and cross-linking with ethylenediamine in a w/o inverse microemulsion. In vitro MRI study using magnetic dextran nanogel and a clinical 1.5 T MRI scanner showed relatively higher T_2_ relaxivity (277.2 mM_Fe_^−1^ · s^−1^) than single Fe_3_O_4_ nanoparticles (5.6 fold), attributed, according to the authors, to the iron oxide cores trapping into the hydrogel network, leading to a lower water diffusion coefficient and increasing of the transverse relaxation rates.

A recent original article by Shin et al. [99] described the synthesis of a nanoparticle with a polysaccharide supramolecular core and a shell of amorphous ferric oxide. The article reported MRI evaluation of cerebral, coronary and peripheral microvessels in rodents and lower-extremity vessels in rabbits. The schematic of the nanoparticle production included the following steps: supramolecular dextran core (synthesized by cross-linking dextran with epichlorohydrin and ethylenediamine) and iron oxide surface coating under basic conditions, both at room temperature. After intravenous administration on BALB/c mice and Sprague–Dawley rats’ tails and rabbit ears, the supramolecular iron oxide generated a strong T_1_ MRI contrast effect, with a relaxivity coefficient ratio of ~1.2. This was close to the ideal value and similar to the gadolinium, using a 3 T clinical MRI system. The authors compared the imaging capabilities with Dotarem (gadoterate meglumine), a clinically approved gadolinium-based MRI contrast agent. The high-resolution T_1_ feature was attributed to the hydrous ferric oxide shell interaction with water molecules. Another important feature of this compound was its excellent in vivo renal clearance, long blood circulation (attributed to 5 nm size and optimized charge) and the fact that it did not accumulate in the organs. All of these serve as favourable evidence of its potential to serve as a contrast agent for cerebral, peripheral and coronary vessels.

#### 3.1.3. Heparin

Heparin is a natural glycosaminoglycan molecule containing sulphate and carboxylic groups, widely used as a clinical anticoagulant and in drug delivery and tissue engineering to improve the blood compatibility of biomaterials. Heparin consists of a complex combination of linear anionic polysaccharides, with an average molecular weight of 16 kDa [169]. It is composed of disaccharide repeating units of d-glucosamine, d-glucoronic, and L-iduronic acid that carry *O*-sulfo, *N*-acetyl or *N*-sulfo groups, resulting in a heterogenic mixture of sulfonated molecules. This high content of anionic groups in the heparin molecule warrants multiple point moieties to the iron oxide nanoparticles surface, mainly through electrostatic interactions.

In 2011, Yuk et al. [95] designed glycol chitosan/heparin-immobilized iron oxide nanoparticles as an MRI agent with a tumour-targeting feature. First, iron oxide nanoseeds were prepared by alkaline co-precipitation of ferrous (Fe^2+^) and ferric (Fe^3+^) chlorides followed by gold deposition through the reduction of Au^3+^ on the iron oxides surface. The next step involved the incorporation of gold-deposited iron oxide nanoparticles into the glycol chitosan/heparin network in the presence of Tween 80 to form composite nanoparticles. In vivo MR images of the composite nanoparticles containing iron oxide showed short spin-spin relaxation times (T_2_*) after dephasing the spin of neighbouring water protons, resulting in the darkening of T_2_*-weighted images, when compared with Resovist. The accumulation in the tumour site even 6 h post administration in C_3_H/HeN mice, indicated that the obtained composites might be utilized as an MR imaging agent with enhanced targeting features.

Heparin-coated SPIONs were developed as a potential T_2_ contrast agent by Lee (2011) [122] and compared with dextran-coated SPIONs, commercially available as Feridex. In the relaxivity measurements, using a clinical 1.5 T MRI system in agarose phantom, heparin-coated SPIO (r_1_ and r_2_ was 9.4 and 170.7 mM^−1^·s^−1^) demonstrated better signals than dextran-coated SPIO (r_1_ and r_2_ was 2.2 and 72.4 mM^−1^·s^−1^), indicating that both could be useful for T_2_-weighted MR imaging due to their high r_2_/r_1_ ratio (above 10). Interestingly, the in vitro cellular labelling results confirmed that heparinized SPIO could visualize rat Ins-1 pancreatic β-cells after a short incubation time (2 h), because of its low r_2_/r_1_ ratio. Once more, from these findings, heparin-coated SPIO can be used as a good negative contrast agent in clinical MRI. The same group tested SPIO nanoparticles coated with unfractionated heparin as new negative contrast agents for in vivo MR imaging of human mesenchymal stem cells (hMSCs) in 2012 [123]. The in vitro T_2_-weighted MRI, performed using hMSCs suspended in low-melting agarose, showed a linear dependency of the signal from grey to dark black with the increasing of coated nanoparticle concentration. The authors also investigated the long-term in vivo tracking of heparin-SPIO labelled hMSCs, by transplanting the cells under the renal subcapsular membranes of the left kidneys of nude mice and monitoring the hypointensity signals with T_2_- or T_2_*-weighted MRI on days 1, 14, and 28. The in vivo results showed strong negative intensity with a detectable range of spatial resolution 28 days after transplantation—results consistent with the histological analysis of the ex vivo kidneys.

In 2017, iron oxide nanoparticles were coated with different heparins of distinct anticoagulant/anti-heparanase activity ratios, and investigated as positive contrast agents in MRI [114]. The authors used a one-step microwave-assisted method for the synthesis of heparin coated-iron oxide nanoparticles with hydrodynamic sizes between 30 and 60 nm. The MRI performances of the obtained materials were assessed by investigating longitudinal (r_1_) and transversal (r_2_) relaxivities at 37 °C and 1.5 T, and the results displayed a r_2_/r_1_ ratio suitable for T_1_-weighted MRI. The in vivo magnetic resonance angiography (7 T) showed a bright vascular architecture after injection in mice tail veins of heparin-coated iron oxide nanoparticles, with excellent anatomical details of carotids, aorta, heart chambers, main veins, and even some smaller vessels, demonstrating that the nanoparticles could enhance T_1_ relaxation in the circulating system.

In 2019, Xie et al. [145] demonstrated that SPIONs coated with a monolayer of succinylated heparin exhibited over four-fold increased T_2_ relaxivity (460 mM^−1^·s^−1^) as compared to Feridex (98.3 mM^−1^·s^−1^) on in vitro MRI relaxivity measurements using a 3.0 T MR scanner. MRI imaging, performed using mice bearing human head and neck tumour (KB cell line) cells with a 7.0 T MRI scanner, confirmed the enhanced T_2_ imaging ability of succinylated heparin monolayer SPIONs (stable 14 nm nanoparticles) and the consequent accumulation in the tumour site—an ability imparted by the thinner coating. It is our belief that the aforementioned results demonstrate the potential of heparin-coated iron oxide nanoparticles as possible high-performance clinical T_2_ contrast agents.

Recently, the same group of authors reported a new magnetic iron oxide nanoparticle with a succinylated heparin monolayer coating. It exhibited the highest T_1_ relaxivity and the lowest r_2_/r_1_ ratio found to date [108]. Magnetic iron oxide nanoparticles with diameters of 2, 3, and 5 nm were synthesized via the thermal decomposition of the iron-oleate complex, then surface-functionalized with succinylated heparin by ligand exchange, obtaining nanoparticles with core sizes ranging from 2 to 18.5 nm. After in vitro and in vivo MRI relaxivity measurements for all size particles, the authors demonstrated that the 2 nm succinylated heparin magnetic iron oxide nanoparticle had a high T_1_ relaxivity of 4.6 mM^−1^·s^−1^ and a low r_2_/r_1_ ratio of 4.0 at 7 T field strength.

#### 3.1.4. Albumin

Human serum albumin (HSA) is the most abundant protein found in the blood. It has excellent biocompatibility as a nanoparticle coating for biomedical applications. Commercially, albumins are obtained from egg white, bovine serum, human serum, milk, grains, and soybeans. Among these, bovine serum albumin (BSA) is the most commonly studied negatively-charged plasma protein, due to the similarity of sequence and structure between BSA and HSA, as well as its non-toxicity, good biocompatibility, and excellent biodegradability, serving often as a stabilizing agent for nanoparticles.

Since 1987, HSA-covered magnetite microspheres have been produced through a modified w/o emulsion polymerization method, as reported by Widder et al. [87], and tested as a contrast agent for MRI, with effects on both T_1_ and T_2_. In following years, multiple studies described different contrast agents based on iron oxide-albumin shells, with the potential to be introduced into clinical imaging (conclusions briefly detailed below).

Hybrid nanoclusters based on BSA/SPION were synthesized in a two-step procedure for liver-specific MRI by Zhang et al. [85]. First, the hydrophobic magnetic nanoparticles were obtained by thermo-chemical decomposition at 300 °C, followed by the formation of oil-in-water emulsion via ultrasonication in the presence of SPION/chloroform organic solution as the oil phase and BSA solution as the water phase. The as-prepared water-soluble BSA/SPION hybrid nanoclusters had a uniform size of ~86 nm, exhibited superparamagnetic behaviour and good colloidal stability and biocompatibility. The liver MRI acquired in vivo after the intravenous injections showed a high value of r_2_/r_1_ ratio (139.7), favourable for T_2_ relaxation enhancement.

Copolymeric micelles—based on poly (2,2,3,4,4,4-hexafluorobutyl methacrylate-co-methacryloxyethyl trimethyl ammonium chloride)-g-methoxy PEG- monomethacrylate functionalized with folate, conjugated with BSA and loaded with SPIONs—were prepared by self-assembling/electrostatic interactions and investigated as a specific contrast agent for tumour targeting and MRI in vitro and in vivo by Li et al. in 2015 [81]. The high T_2_ relaxivity was demonstrated using the obtained micelles for in vitro studies and sustained by in vivo tumour-specific MR imaging of hepatoma at 24 h post injection. The functionalized micelles were internalized by the tumour, as observed by the in vitro cellular uptake studies and prolonged circulation time, revealing their potential as a tumour-targeting contrast agent.

Superparamagnetic iron oxide, this time with γ-Fe_2_O_3_ nanoparticles, were synthesized from Fe_3_O_4_, then surface-coated with BSA, conjugated with tumour-specific ligand folic acid and decorated with FITC for dual-modal imaging (MRI and intracellular internalization and visualization) in human brain tumours [91]. In vitro MRI assessment on a 1.5 T scanner produced a significantly negative contrasted T_2_-weighted MR phantom imaging (signal darkening) of the in-human brain tumour U251 cells. The fluorescent capability was also demonstrated by intracellular internalization within the U251 cells treated with the obtained material.

Tzameret et al. [117] prepared core–shell near infrared fluorescent (NHS Cy7) iron oxide nanoparticles coated with BSA shell and tested in vivo tracking (by MRI) into the posterior segment of the eye in a rat model of retinal degeneration. The bioactive magnetic iron oxide/HSA nanoparticles were detected in the back part of the rats’ eyes by MRI for up to 30 weeks following injection, with effects on T_2_*—and were detected for up to 6 weeks by histology, suggesting the nanoparticles’ potential use for extended release drug delivery in the posterior segment.

A comparison of the T_1_-weight contrast performances of iron oxide nanoparticles modified with BSA and poly (acrylic acid)-poly(methacrylic acid macromolecule ligands with similar size and magnetization was reported by Tao et al. [80] in 2019. The obtained nanoparticles, coated with BSA, exhibited higher r_2_/r_1_ ratios in solution and darkening contrast enhancement for liver and kidney sites of mice under T_1_-weight imaging on a 0.5 T MRI scanner, when compared with the artificial macromolecule coated nanoparticles. Additionally, the Fe_3_O_4_-BSA displayed T_2_ contrast enhancement, demonstrating the dependence of MRI performance on the nanoparticle surface.

In a recent study, BSA nanocage protein was used as a biotemplate to synthesize ~3.5 nm uniform monodispersed Fe_2_O_3_-BSA nanoparticles with good biocompatibility and a high T_1_ contrast effect [90]. The authors selected Fe_2_O_3_ instead of the more common Fe_3_O_4_ because it was reported that the first type of iron oxide increases the value of r_1_, maximizing the T_1_ contrast effect and weakening the T_2_ contrast effect [170]. The in vitro MRI relaxivity study, acquired on a 3 T scanner, showed a factor of 1.8 higher for r_1_ values when compared with the commercially available ferumoxytol and gadolinium-diethylenetriamine pentaacetic acid (Gd-DTPA) values. In vivo MRI study was performed in healthy Sprague Dawley rats, and the same 3 T scanner demonstrated the clinical potential of the self-assembled BSA nanocages as T_1_-weighted contrast agents. The highest positive contrast appeared in the heart 15 m after injection.

#### 3.1.5. Gelatin

Gelatin is a naturally-derived macromolecule with a high hydration degree, high biocompatibility and biodegradability, available at a low market price. It is obtained through the partial hydrolysis of native collagen, both acidic (for gelatin type A) and alkaline (for type B). Gelatin has been exploited in the biomedical field as a drug carrier and/or contrast agent, as well as a cell culture substrate in a variety of forms, owing to its unique chemical and physical nature. According to Narayanaswamy et al. [57], there have been very few studies focused on the design and manufacture of gelatin-based iron oxide nanoparticles as imaging agents (as of 2014, briefly mentioned in Table 4). Below, we indicate several recently reported findings that fit the purposes of this review, after an ample literature search performed using the PubMed, Web of Science and Scopus databases using the following terms: ”gelatin” AND ”iron oxide” AND ”imaging—a search that returned 47 studies.

A multimodal probe capable of visualizing cells by optical and in vitro MRI modalities was prepared by emulsion method. It consisted of gelatin nanospheres incorporating quantum dots and iron oxide nanoparticles, cross-linked with glutaraldehyde and treated with octa-arginine (R8) of a cell-penetrating peptide [112]. The composite nanospheres (about 162 nm) were efficiently internalized into the cells, as visualized by both confocal laser microscopy and T_2_-weighted MRI modalities.

In a more recent study, the current 3D printing method was employed for repairing and replacing diseased bile ducts using an artificial tubular composite scaffold based on polycaprolactone (PCL) as a matrix for the organoid cells of the bile duct [151]. In order to enhance the scaffold features, a layer of gelatin methacryloyl (GelMA) hydrogel was applied on the outer layer, followed by USPIO nanoparticle dispersion for contrast agent potential. The T_2_-weighted in vitro MRI images of the scaffolds using a clinical 3 T full body scanner illustrated a uniform signal area with clear continuous boundaries, which could clearly display position changes and degradation of the scaffold in real-time.

#### 3.1.6. Alginate

Alginate is a linear anionic polysaccharide usually extracted from brown algae and formed by α-l-guluronate (G) and β-D-mannuronate (M) copolymers arranged in a block structure as a homopolymer (consecutive poly-G/poly-M residues) or heteropolymer (an alternating M and G residues). Alginate has been extensively investigated and used for many biomedical applications due to its hydrophilicity, biocompatibility, biodegradability, lack of toxicity, mucoadhesiveness, relatively low cost, pH sensitivity, mild gelation with the addition of divalent cations, and lack of immunogenicity—having already received permission from the FDA for human use [171].

Bar-Shir et al. [143] used both in vitro and in vivo MRI studies for non-invasive determination of the Ca^2+^ level changes extracellular and in deep tissues, using selective SPIONs coated with monodispersed and purified alginate. The added Ca^2+^ sensors (obtained from ischemic astrocyte cell culture in DMEM) were detected by T_2_-weighted MRI—since the nanoparticles aggregated in the presence of Ca^2+^—and compared with images of the reference solutions. The alginate-coated magnetic nanoparticles were further tested in vivo in a quinolinic acid model of neurotoxicity in rat brains as proof of concept at 7 or 12 days post injection. The T_2_-weighted MR images clearly showed that the nanoparticles could be qualitatively used as a platform for the non-invasive MRI determination of extracellular Ca^2+^ levels.

Iron oxide nanoparticle-immobilized alginate nanogels were reported as novel contrast agents for enhanced MR imaging applications. Briefly, alginate carboxyl groups were activated by 1-ethyl-3-(3-dimethylaminopropyl) carbodiimide hydrochloride, followed by double water-in-oil emulsion formation using dichloromethane, dioctyl sodium sulfosuccinate and aqueous PVA solution to synthesize the nanogels, and in situ cross-linking with PEI-coated Fe_3_O_4_ nanoparticles obtained by the hydrothermal method [86]. The obtained alginate/PEI-Fe_3_O_4_ nanogels (with a size of 186.1 nm) were water-dispersible, colloidal with high stability, and cytocompatible in the given concentration range. As suspected, the nanogels showed a negative contrast for T_2_-weighted MR imaging with a high r_2_ relaxivity (170.87 mM^−1^·s^−1^) as observed in vitro on cancer cells and in the xenografted tumour model in vivo after intravenous injection.

### 3.2. Synthetic Biocompatible Polymers

The unique identities and features of these polymers’ architecture makes them suitable for use in the design of versatile, polymeric functionalities and demonstrates their potential use in specific and specialized applications. The term biocompatible has attracted significant attention in the biomedical field, wherein the introduction of specific polymeric moieties on a system defines extraordinary functions compatible with biological systems [172].

Over the years, developments in synthetic strategies for iron oxide nanoparticle coatings (e.g., PEG [173,174], PVA [175], PLGA [176], poly (*N*-isopropylacrylamide) [177], etc.), as presented in Table 5, have defined multiple vehicles for diagnostic applications. Therefore, we described only silica and PEGylation strategies in this section, since our group has contributed to this field.

#### 3.2.1. Silica Shells

Silica and its derivatives offer high-quality, chemically stable, optically transparent, nontoxic and biocompatible shells to iron oxide cores. This tailored surface improves the nanoparticle dispersion in aqueous saline media, reduces aggregation and offers an excellent platform to attach various biological ligands like proteins, nucleic acids, etc. via silane linkers.

Magnetic iron oxide nanoparticles coated with silica or alkoxysilanes are currently being screened as contrast agents by numerous research groups, as presented in Table 5, demonstrating their efficacy and their dependence on composition, size and shape.

In 2004, Yan et al. [223] synthesized silica-embedded iron oxide nanoparticles via sol-gel hydrolysis as a new approach for MRI contrast agent design. The 220 nm nanoparticles were evaluated on a 7 T scanner. The in vitro results showed reduction in signal intensity in the T_2_-weighted images, demonstrating their potential as an effective MRI contrast agent carrier. Following this study—considered the first report to have used silica-coated iron oxide nanoparticles as a contrast agent in MRI—in 2007, Zhang et al. [260] studied the ability of silica- and alkoxysilane-coated USPIO particles to label immortalized progenitor cells for MRI. First, the authors synthesized USPIO particles by co-precipitation of ferric and ferrous salts, followed by three types of coatings: silica (using sodium metasilicate pentahydrate), APTMS, and AEAPTMS. Thereafter, starting from nearly 9.6 nm for USPIO particles, the authors obtained particles of 9.9 (silica), 10.5 (APTMS) and 10.9 nm (AEAPTMS) after the surface coating. MR measurements performed with cell pellets showed high T_2_ relaxivities for all particles and cell concentrations and were intensely internalized in immortalized progenitor cells, making them suitable for MRI cell labelling.

A more complex approach to developing a promising vehicle for MR imaging was reported in 2008 by Feng et al. [181]. The authors synthesized magnetite nanoparticles, coated with APTES using a silanization reaction and then linked with PEG diacid via covalent bonds. The well-dispersed surface-functionalized biocompatible-functionalized magnetic nanoparticles had an average size of 20 nm and exhibited superparamagnetism. They were used to perform MRI experiments on living rabbits with VX2 malignant tumours. The signal decreases in the tumours after intravenous administration of the magnetic vehicles indicated the accumulation in malignant tumour tissues, as presented in the T_2_*-weighted MR images, demonstrating the potential of the authors’ nanoparticles to be used in MR imaging.

A multifunctional imaging probe for MRI was produced in 2010 [255] by combining SPIOs, a porous silica shell and Cu^2+^ for cell labelling. Highly crystalline and non-aggregated SPIOs, either single or clustered, were encapsulated within the silica matrix, providing stability, biocompatibility and a high surface area that could be easily functionalized with different ligands. It is important to mention that the authors developed a unique ligand based on EDTA-bis (3-triethoxysilyl-*n*-propyl amide) which exposed the dependence of the nanoparticles’ endocytosis and uptake on the surface charge. Following the relaxivity measurements through the dispersion of the nanoparticles in 0.1% agarose gel, it was evident that the clustered nanoparticles provided a significantly enhanced T_2_ relaxivity in comparison to single SPIOs, comparable to commercially available Feridex. The presence of copper on the surface of the nanoparticles reduced the r_2_ values but exhibited high cell uptake efficiency, which is useful for cell tracking for the detection of Wilson’s disease.

Campbell et al. [264] evaluated quasi-cubic iron oxide/silica nanoparticles of sub-100 nm size as T_2_ contrast agents for MRI of biological tissues. The authors noted that the silica coating improved the core-shell stability on long-term storage conditions, when compared with commercially available MRI contrast agents. In vitro cell studies and cytotoxicity assays on human prostate cancer cells (PC3 cell line) demonstrated that their uptake was more efficient than a similar concentration of bare magnetite nanoparticles. The relaxivity measurements were performed in phantoms, after being uptaken by PC3 prostate cancer cells, on a 3 T clinical MRI scanner, which exposed their capability as T_2_ MR contrast agents, as expected. This was mainly due to their relatively high relaxivity and saturation magnetization values. The preliminary in vivo MRI studies in a breast tumour mouse model also sustained the T_2_ signal enhancement at the tumour site post injection of the quasi-cubic magnetite/silica core-shell nanoparticles. As the authors mentioned in the conclusions section, future studies on this type of nanomaterial could be designed to accurately diagnose pathologies.

A dual-contrast T_1_- and T_2_-weighted MRI agent of about 21 nm was designed by Yang et al. [183], comprising a superparamagnetic iron oxide core synthesized via a thermal decomposition approach, a silica shell aminated through silanization, gadolinium complex (Gd-DTPA) and an arginine-glycine-aspartic acid peptide as a targeting ligand, covalently conjugated onto the surface. Relaxivity measurements—along with the in vitro and in vivo MR imaging performed with a 3.0 T system—exhibited dual-contrast ability with a high degree of accuracy from the multifunctional Gd-labelled SPIONs for targeted imaging of a tumour model on a nude mouse.

A typical procedure for the synthesis of uniform mesoporous silica-coated iron oxide nanoparticles, e.g., thermal decomposition of Fe-oleate complex followed by silica coating via a surfactant-templated sol–gel method, was proposed by Ye et al. [204]. Different silica shell thicknesses were prepared in order to evaluate the effect of surface coating on MRI contrast efficiency. Results demonstrated a significant impact; thick layers enhanced the MRI contrast, results which were also demonstrated by NMR relaxometry studies. The biocompatible nanoparticles were examined on epithelial cells derived from the organ of Corti of transgenic mice, and displayed enhanced in vitro MRI efficiency in T_2_ sequence, up to 21 times higher than commercial agents. These results supported the potential application of the material as a highly efficient MRI T_2_ contrast agent.

The same concept, exploring the effects of silica shell thickness of iron oxide nanostructures on MRI contrast, was adopted by Joshi et al. [202] in the same year. In this paper, the authors prepared 9 nm Fe_3_O_4_ in organic phase via a simple chemical decomposition method with 5, 10, and 13 nm silica shell thickness coatings by base-catalysed silica formation from tetraorthosilicate in a reverse micro-emulsion procedure. The multiple-echo-spin-echo sequence scans on a 3.0 T MR system showed that the increase in thickness of silica shells in core-shell nanostructures produced a decrease in r_2_ relaxivity. This phenomenon was attributed to the long distance between the magnetic core and water molecules.

More complex architectures were developed over the years. A combined MRI/PET, easily-prepared agent that offered signal or contrast in both modalities was described by Burke et al. [225]. The new class of silica-coated iron oxide nanorods were coated with PEG and/or a tetraazamacrocyclic chelator (DO3A), and were developed by the same group in 2014 [224]. They were evaluated as in vivo T_2_ MRI and PET contrast agents. The magnetic behaviour investigated in vitro at 3 T showed that all three types of nanorod constructs had fast relaxivity as T_2_ contrast agents, but were relatively weak as T_1_. The in vivo imaging biodistribution and stability on PET-CT and MR imaging of gallium-68 radiolabelled nanorods displayed the expected high liver uptake with no significant release of the positron-emitting radioisotope metal. The results validated the novel method for chelator-free radiometal labelling of silica-coated iron oxide nanorods via surface interactions—that could be used for high-sensitivity liver imaging.

A classic route—co-precipitation and Stöber methods, followed by mebrofenin functionalization—was proposed by Yazdani et al. [184] in their work to produce a liver-targeting MRI contrast agent. Their in vitro studies (nanoparticles dispersed in water) on a 1.5 T MRI scanner showed the effects of iron concentration on relaxivity values (r_2_); namely, that the T_2_-weighted signal decreased with increasing Fe concentration. More studies are needed to demonstrate the ability of the added liver targeting function to provide contrast to the envisioned organ, as we could not find any additional studies by the authors.

New silica-coated cubic SPIONs, synthesized using the thermal decomposition method, showed a synergistic T_1_- and T_2_-contrast-enhancement for MRI on both in vitro (on phantom vials and L929 line cell) and in vivo studies (post injection on the tail vein of Sprague Dawley rats) [253]. The authors compared the relaxation performance of cubic SPIONs with their spherical counterparts of different sizes (7, 11 and 14 nm) on a 3 T MRI scanner, demonstrating that, when controlling the shape and size of the nanoparticles, the relaxivity values differed. Promising results were obtained for the 11 nm silica-coated cubic SPIONs, which were determined to be potential candidates for dual-mode contrast agents.

An alternative to the toxic Gd-based contrast agents used in MRI was proposed by the Mathieu team, who introduced the iron–iron oxide core-shell architectures, obtained as follows: synthesis of 10 nm large iron–iron oxide nanoparticles, silica coating with 11 nm thick layer by reverse emulsion method and functionalization with 5 kDa PEG chains [238]. The nanoparticles produced were in the 100 nm range, displayed good stability in water, no cytotoxicity, high r_2_ relaxivity values and low r_1_, leading to enhanced r_2_/r_1_ ratios in comparison with commercially available Resovist, as observed on T_2_ MRI phantoms images acquired on 9.4 T scanner.

The ultrafine silica-coated superparamagnetic iron oxide fluorescent nanoparticles prepared in 2017 [265] were evaluated in 2019 [185] for their biocompatibility and biosafety profile when used to label human amniotic mesenchymal stromal/stem cells (hAMSCs) or when administered in vivo, along with in vitro magnetic responsiveness. Briefly, the synthesis involved the classic co-precipitation strategy, followed by one-pot synthetic procedure for thin silica shell coating and FITC covalent attachment. The obtained nanomaterials showed excellent cytocompatibility and were internalized with no interferences on the stem cells’ characteristics on in vitro tests. The 7-week in vivo study in mice indicated the biocompatibility and biosafety profile over short and long term periods with a biodistribution dependent on time, demonstrating the potential for this material to be used as an excellent MRI tracking agent.

Another prospective study with potential in imaging was found in 2020 by Navarro–Palomares et al. [210]. Their paper showed the development of two-type multifunctional nanoparticles based on hydrophobic Fe_3_O_4_ (prepared by hydrothermal approach) and commercial ZnO, both coated with SiO_2_ incorporating rhodamine B isothiocyanate, fluorescein or rhodamine B, following either a reverse microemulsion route or a modified Stöber method, respectively. The in vitro studies on HeLa cell lines confirmed that the silica shell conferred stability and biocompatibility and could be degraded in different physiological media. These are relevant criteria for potential applications in biomedicine and related fields. No further studies related to imaging were found in the literature using the above-mentioned dye-doped biodegradable nanoparticle SiO_2_ coating in zinc- and iron oxide nanoparticles.

A dual MR/NIRF imaging agent, used to identify macrophage enrichment in atherosclerotic plaques, was successfully synthesized in several steps: iron oxide nanoparticles (produced by thermal decomposition approach), mesoporous silica deposition, PP1 peptide conjugation via the amide condensation reaction, and NIRF dye (IR820) loading [232]. The core/shell nanoparticles had a uniform size of 90 nm, were internalized by active foamy macrophages and favoured atherosclerotic plaque imaging in MR/NIRF dual-modal (combined T_2_ and T_2_^*^ mapping on 3 T equipment) on atherosclerosis models of ApoE^−/−^ mice.

#### 3.2.2. PEG

PEG is a typical nontoxic, nonantigenic, coiled polymer composed of repeating ethylene ether units with dynamic conformations [266,267]. It is inexpensive, versatile and FDA approved for many applications. It exhibits many useful features, e.g., hydrophilicity, biocompatibility and the capacity to enhance nanoparticles’ blood half-time. The PEG coating, named PEGylation, boosts stability in an aqueous medium, prevents particle surfaces from oxidizing, reduces toxicity, and lowers RES uptake, thus increasing circulation time.

In imaging, PEG moieties are added in the iron oxide nanoparticle reaction or in a subsequent step, as mentioned above for other polymers, in order to satisfy the contrast agent specifications: dispersibility in aqueous media, monodispersity, biocompatibility and so on.

Numerous studies were found in the literature that dealt with PEG coating, functionalization or attachment to the iron oxide core, starting from 1997. Below, we have highlighted a few of the achievements needed to reach preclinical and/or clinical use. More studies focused on PEGylated anchoring groups are presented in Table 5.

In 2008, Park et al. [268] carried out one-step synthesis of PEG surface-modified USPIONs in a polar organic solvent and quantified them with 1.5 T MRI in solution. The T_1_ and T_2_ map images showed a low r_2_/r_1_ relaxivity ratio with a clear dose-dependent effect, indicating that the as-prepared nanoparticles could be considered potential MRI contrast agents for both T_1_ and T_2_ sequences.

A PEG polymer containing a bisphosphonate anchor covered USPIOs surface using a simple method of 1 h at room temperature [261]. The resulting nanoparticles revealed a high r_1_ of 9.5 mM^−1^·s^−1^ and low r_2_/r_1_ ratio of 2.97, suitable for T_1_-weighted MRI contrast (results obtained in vitro at 3 T). The PEGylated nanoparticles were injected into the tail veins of BALB/c mice and therein enhanced the signal from blood vessels and vascular systems with minor accumulation in the liver. This reinforced their potential use as a contrast agent for T_1_-MRI angiography. Furthermore, the authors tested them after radiolabelling with gamma-emitting isotopes (^99m^Tc) for biodistribution in vivo using SPECT imaging. The studies confirmed low RES uptake and long blood circulation times, validating their potential use as a dual-modality imaging agent.

Yang et al. [263] reported a simple one-pot reaction for highly water-stable iron oxide nanoparticles synthesis in 2014, using PEG as solvent, capping and reducing agent. The authors tried to develop multimodal-imaging agents. The USPIONs were amine-functionalized using dicyclohexylcarbodiimide/*N*-hydroxysuccinimide coupling, followed by fluorescein isothiocyanate labelling. The T_2_* coronal MR and fluorescence imaging performed on BALB/c nude mice demonstrated that the simply-synthesized nanoparticles showed the potential to be used for in vivo multimodal imaging.

Functional magnetic nanoparticles, modified with PLA-PEG-D-glucosamine that showed good biocompatibility and stability both in vitro on RAW 264.7 macrophages and 4 T1 cell line and in vivo on tumour-bearing BABL/c mice, were successfully evaluated as MRI contrast agents for tumour imaging [182]. The authors synthesized superparamagnetic Fe_3_O_4_ nanoparticles capped with oleic acid via the co-precipitation method and PLA-PEG-D-glucosamine complex, followed by their linkage through hydrophobic interactions. After demonstrating the ability of covalent-linked D-glucosamine on the nanoparticle surface to increase the specific uptake of tumour cells in vitro, the nanomaterials were injected via tail vein in a low dose, and their MRI performances were compared with Resovist. At 2 h post-injection, the multifunctional nanoparticles accumulated in tumour tissue, as observed on T_2_* weighted images.

Multicore iron oxide nanoparticles (maghemite) coated with poly (4-vinylpyridine)-PEG copolymer and having an average hydrodynamic diameter of approximately 20 nm, crossed the glomerulus wall and were mostly excreted through the urine. Thus, they avoided the effects of RES, as confirmed by SPECT, gamma counting (after radiolabelling with ^111^In ions), T_2_-MRI biodistribution studies on BALB/cJRj mice. The simple, new, reliable and direct radiolabelling method presented by the authors could be applied to prepare magnetic nanocarrier MRI/SPECT contrast agents for kidneys [222].

A promising T_1_-T_2_ dual-mode contrast agent for MRI was developed in a recent study, consisting of superparamagnetic manganese oxide-doped iron oxide (Fe_3_O_4_/MnO) nanoparticles of approximately 20 nm stabilized with hydroxyl− PEG-phosphonic acid [199]. The T_1_-T_2_ weighted images—obtained using a 0.5 T MRI scanner—validated the dual-mode contrast ability on phantoms. In vivo MR imaging on BALB/c mice using 7 T MRI equipment confirmed the positive signal enhancement and remarkable T_1_-T_2_ relaxivity of decorated superparamagnetic nanoparticles.

Tailor-made PEG-coated iron oxide nanoparticles were synthesized in a two-step reaction; first by solvothermal synthesis in organic medium, followed by surface modification with different molecular-weighted PEGs through amide bond formation. After a detailed physicochemical characterization, the 100 nm nanoparticles showed in vitro biocompatibility and in vivo safety, high cell uptake in tumoral cells and the highest r_2_ relaxivity values when compared with commercial Ferumoxytol, promoting their applicability in MRI on mice bearing xenografted human breast cancer models [235].

Among various types of fabricated nanoparticles, the Karahaliloglu team [221] recently reported PEG-terminated, PS-linoleic copolymer-coated magnetic iron oxide nanoparticles as unique prospective candidates for diagnosis of hepatocellular carcinomas. Magnetic iron oxide nanoparticles were synthesized by the co-precipitation method and subsequently incorporated into the previously prepared PEG-terminated PS-linoleic copolymer core. The coated magnetic samples were scanned under a 3 T MRI scanner in an aqueous solution and showed good T_2_-weighted contrast at various concentrations, with signal intensity dependent on the iron oxide concentration.

## 4. Imaging Pre-Clinical and Clinical Studies of Core-Shell Iron Oxide Agents

The challenges for clinical translation start from the design of the preparation method and continue up until preclinical evaluation as one pillar and the ultimate goal, use in clinical practice, as the other pillar. As we can observe from the above sections, animal molecular imaging has become an efficient tool to understand and clarify the biological mechanisms for accurate diagnosis. Over the past 10 years, the two fields (preclinical and clinical) have begun to overlap more and more. The enormous progress made toward development of related imaging technologies has guided researchers down the road toward the already-available iron oxide nanoparticle-based contrast agents [269].

Table 6 points out just a few examples of nanomaterial tracers that have been assessed in numerous preclinical studies, ranging from breast and cardiac to brain monitoring. Notably, biocompatible iron oxide-based nanoparticles that have been approved by the FDA for clinical use as MRI contrast agents (as presented in Table 7) are still paving the road toward achievement of the critical requirements for use in practical applications [270]. Their chemical stability, dispersion in biological media, uniformity in size and diverse coatings continue to make them the subject of numerous articles and studies in ongoing efforts to develop new candidates.

Although several nanoparticles with iron cores have been approved for human use, gaps in technical knowledge, capabilities and the lack of the safe-by-design mindset in many groups continue to prevent them from progressing from the bench to the bedside. This is the current situation, since the first nanoparticle-based iron oxide imaging agent, Feridex I.V.^®^ (ferumoxides) was approved by the FDA in 1996 for the detection of liver lesions. According to Tassa et al. [278] in 2011, Combidex^®^ (ferumoxtran-10) was the next approved agent, and was used for prostate cancer lymph-node metastases imaging. That was followed by Feraheme^®^ (ferumoxytol), dedicated to treating iron deficiency anaemia in adult patients with chronic kidney disease and the detection of inflammation of the central nervous system, brain neoplasms and cerebral metastases from lung or breast cancer (still under clinical investigation).

According to FDA indications, there are critical facts about ferumoxytol, including fatal outcomes upon administration, serious allergic reactions, hypotension, and iron overload that could lead to death. Based on these side effects, the FDA gave strict warnings regarding potentially serious hypersensitivity/anaphylaxis reactions. Fatal events have occurred in patients receiving ferumoxytol; the initial symptoms were commonly hypotension, syncope, unresponsiveness, and cardiac/cardiorespiratory arrest [279].

In the same year, 2011, Wang [280] described SPIO-based MRI contrast agents, i.e., ferumoxides (Feridex in the USA, Endorem in Europe, also referred as AMI-25), ferucarbotran (Resovist, SH U 555A), ferumoxtran-10 (AMI-227, Combidex, Sinerem, Guerbet), Clariscan (PEG-fero; Feruglose; NC100150) and iron oxide-based agents for gastrointestinal contrast: AMI-121 (Ferumoxsil, Lumirem, Gastromark) and OMP (Abdoscan). The two clinically-approved SPIONs—defined as a conglomerate of numerous nano-sized iron oxide crystals coated with dextran or carboxydextran, e.g., ferumoxides and ferucarbotran—were specifically appropriate for MR imaging of the liver. Following intravenous administration, their pathway included clearing from the blood by phagocytosis by RES, intracellular uptake, metabolization and integration as ferritin/haemoglobin. Both SPIO particles produced strong T_2_/T_2_* relaxation effects in the liver tissue with the capacity to differentiate lesions, while the smaller Resovist had an enhanced effect on T_1_-weighted images. The review paper concluded with the perspective that new applications of approved SPIO need to be determined and that new SPIOs with relevant properties need to be developed.

However, ferumoxide and ferucarbotran agents (Endorem, Guerbet, Feridex, Resovist) were withdrawn from the market in 2009 due to multiple side effects. One example of these side effects was that the agents could not be administered as an intravenous bolus without risking the possible appearance of severe backache. Thus, it is imperative to evaluate biosafety before going to market [281]. Additionally, the clinically approved SPIONs were unable to differentiate between hepatocellular carcinoma and healthy liver tissue.

According to the information gathered from the ClinicalTrials.gov database (Table 7), there are currently numerous suitable MRI and MRI/SPECT-based contrast agents under evaluation for use in humans. They are undergoing extensive safety and toxicology studies, as the regulatory body demands, with the ultimate goal of improving patient quality of life [282]. Nonetheless, as numerous studies point out, the final judge of a contrast agent remains the patient that genuinely understands the significance of that experience.

## 5. Authors Contributions

The first step toward the use of iron oxide nanoparticles as a potential contrast agent was in 2019, when the group presented the design of a new multifunctional hybrid magnetic tracer that would be further radiolabelled and used as a dual-modality SPECT and MRI imaging probe at an international conference. Hybrid magnetic nanoparticles were synthesized using iron oxide core and multifunctional silica shell chains available for fluorescent marking and ^99m^Tc radiolabelling. The magnetite nanoparticles were prepared by two wet methods, namely co-precipitation and partial oxidation, in order to compare their functionalization and assembly ability within the polymeric matrix under the influence of the magnetic field. The two types of iron oxide nanoparticles used were described in a published paper in 2015 [283], with minor modifications. The structure, external morphology, size distribution, colloidal and magnetic properties were characterized by FT-IR, TEM, XRD, DLS and VSM analyses. TEM and DLS results showed that the hybrid complex had nanostructures with broad distribution. The formation of crystalline magnetite nanoparticles was confirmed by XRD analysis. Magnetization measurements on the obtained samples showed a straightforward correlation between the saturation magnetization and morphology of the samples. These positive findings suggested that the multifunctional magnetic nanoparticles have the potential to be used in biomedical applications.

Therefore, in the same year, we presented a paper dealing with in vitro cytotoxicity based on MTT assay of magnetic amine-functionalized Fe_3_O_4_ nanoparticle (both types) solution containing BSA, in normal V79 cell line, revealing that the conjugates were noncytotoxic to normal cells.

Furthermore, the amine groups on the hybrid magnetic nanoparticles surface—synthesized using iron oxide cores obtained by the co-precipitation method followed by multifunctional silica shell chain coverage—were radiolabelled with ^99m^Tc, and tested as in vivo imaging agents. A paper was presented at the 32nd Annual Congress of the European Association of Nuclear Medicine and published as an extended abstract in the European Journal of Nuclear Medicine and Molecular Imaging Supplement [284]. Briefly, the magnetic amine-functionalized Fe_3_O_4_ nanoparticles were successfully labelled with ^99m^Tc using standard radiolabelling methods, as revealed by the radiolabelling yield (≈90%) assessed using instant thin layer chromatography. Both the ex vivo biodistribution at 6 h post injection and in vivo SPECT imaging on healthy animals showed a similar pharmacokinetic biodistribution profile for ^99m^Tc-amine Fe_3_O_4_ nanoparticles—but differed from the control batch (^99m^Tc). The hepatic and splenic accumulations of the radiolabelled nanoparticles demonstrated the high uptake in the mononuclear phagocyte system, which was in agreement with the behaviour and clearance of nano-targeted nanoparticles in vivo. In summary, we developed new molecular imaging probes based on hybrid magnetic nanoparticles radiolabelled with the diagnostic radionuclide ^99m^Tc for SPECT imaging and evaluated their biodistribution profiles in healthy animals.

However, our group objective was to develop a multifunctional radiolabelled hybrid magnetic nanoparticle agent for dual-modality (SPECT and MRI) medical imaging. Therefore, in 2020, we prepared and evaluated the toxicity and MRI biodistribution of the same magnetic hybrid nanoparticles. As mentioned above, the hybrid magnetic nanoparticles synthesis (about 21 nm as depicted by TEM images) included the iron oxide core obtained by co-precipitation method and silica shell chain coverage. The in-situ stability of the magnetic hybrid nano-systems was explored over time by size quantification and revealed relative stable nanoparticles in simulated physiological media. The in vivo toxicity studies of the synthesized nanoparticles in healthy animal models over short and long-term periods were assessed by evaluating changes in blood chemistries, variations of blood cell parameters, profiles in liver and kidney or change in gross or histologic features of organs as well as monitoring of clinical and weight changes after intravenous administration. The results exhibited slight variations in Wistar rats during the experimental period, mainly due to an enhanced immune response and inflammatory reactions. Significant MRI signal change was observed over time on liver, aorta, cava vein and hepatic vein on T_2_ sequence biodistribution of the obtained hybrid magnetic nanoparticle probe on healthy experimental animals using 1 T PET/MRI scanner. Furthermore, the quantification of liver iron concentration by signal intensity ratio on MRI demonstrated the ability of the synthesized nanoparticles to act as a new contrast agent. The obtained results present opportunities: to extend this new nanotracer for therapeutic use due to its versatile functionality, or to link different molecules to the same core. Overall, our results suggest that the multifunctional hybrid magnetic nanoparticles could be employed as potential imaging vehicles for targeting tumour tissues.

## 6. Conclusions and Perspectives

Since the first MRI contrast agent based on ferric chloride in 1980, many tracers have been developed and evaluated in vitro, in vivo and also in clinical practice—though some of them were later withdrawn as a result of safety concerns. It is well known that iron oxide-based nanomaterials can enhance diagnostic imaging techniques, and their fusion with molecular imaging agents permit accurate diagnostics at the molecular level. This review points out that a single magnetic nanoparticle can be tagged with various moieties, ligands, imaging agents and/or radionuclides to construct a personalized diagnostic agent characterized by biocompatibility, biodegradability, biosafety, selectivity, stability and controlled biodistribution. However, even after many years of investigation, there are still multiple challenges—from chemical, biological and even economic perspectives—that must be taken into account in translating nanoparticle imaging agents to a clinical setting. Understanding these considerations will define the future of nanoparticle imaging agents that will one day have the capability and functionalization required for personalized diagnostics. 

## Figures and Tables

**Figure 1 molecules-26-03437-f001:**
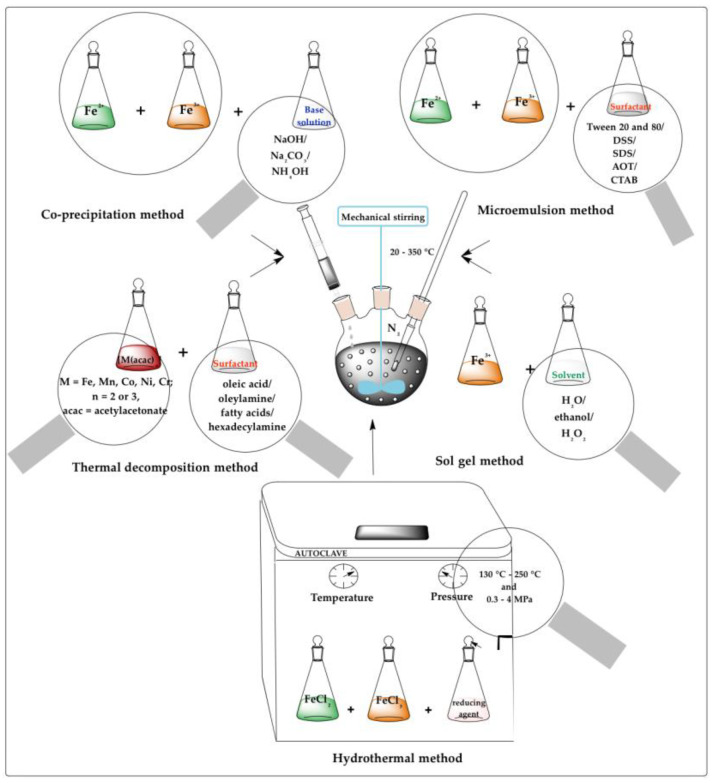
Techniques for synthesis of iron oxide core for imaging purposes.

**Figure 2 molecules-26-03437-f002:**
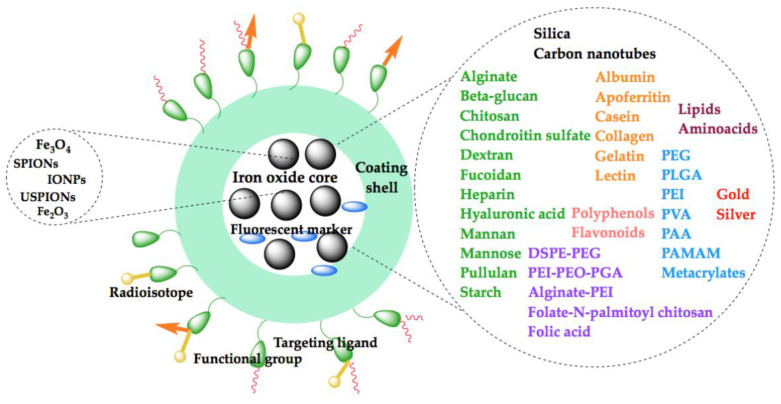
Schematic illustration of the coatings used for iron oxide nanoparticle functionalization for diagnostic purposes.

**Table 1 molecules-26-03437-t001:** Iron oxide-based MRI contrast agents, as described in MICAD [15], with Fe_3_O_4_/SPIONs/USPIONs/iron oxide nanoparticles as a source of signal.

Molecule	Target Region	Level of Research
Lactoferrin-conjugated PEG-coated Fe_3_O_4_ nanoparticles	Lactoferrin receptors	In vitro, rodents
Lactoferrin-conjugated SPIONs		
Anti-ligand–induced binding sites antibody conjugated to microparticles of iron oxide	Platelet glycoprotein GPIIb/IIIa receptor (CD61/CD41)	
Anti-malondialdehyde-modified low-density lipoprotein MDA2 monoclonal antibody–labelled lipid-coated SPIONs	Malondialdehyde-modified low-density lipoprotein (antigen)	
Anti-malondialdehyde-modified low-density lipoprotein MDA2 monoclonal antibody–labelled lipid-coated USPIONs		
Anti-vascular cell adhesion molecule antibody M/K-2.7 and anti-P-selectin antibody RB40.34 conjugated microparticles of iron oxide	VCAM-1 and P-selectin	
Anti-vascular cell adhesion molecule antibody M/K-2.7–conjugated microparticles of iron oxide	VCAM-1	
USPIO-cyclo (Cys-Asn-Asn-Ser-Lys-Ser-His-Thr-Cys)		
Avidin-coated baculoviral vectors-biotinylated USPIONs	-	
CLIO-(H-2Kd)-Lys-Tyr-Asp-Lys-Ala-Asp-Val-Phe-Leu	H-2Kd-restricted β cell-specific T cell receptor	
Complement receptor type 2-conjugated gold/SPIONs	Complement C3 fragments	
Cross-linked iron oxide-transactivator transcription	Adsorptive endocytosis, phagocytosis	
Cyclo(Arg-Gly-Asp-d-Try-Glu) conjugated to USPIONs	Integrin α_v_β3	
FluidMAG iron nanoparticle-labelled mesenchymal stem cells	Cell imaging	
Poly(*N,N*-dimethylacrylamide)-coated maghemite nanoparticles		
Amine-modified silica-coated polyhedral SPIO–labelled rabbit bone marrow–derived mesenchymal stem cells		
Glycol chitosan/heparin-immobilized gold-deposited iron oxide nanoparticles	Fibrinogen-derived product in tumour stroma	
Iron oxide–ferritin nanocages	Non-targeted	
Thiol-modified PEG-conjugated gold/SPIONs		
Magnetic iron microbeads coupled with HEA-125 monoclonal antibody	Epithelial cell adhesion molecule	
H18/7 F(ab’)2 E-selectin monoclonal antibody conjugated to cross-linked iron oxide nanoparticles	E-selectin	
MES-1 F(ab’)2 E-selectin monoclonal antibody conjugated to USPIONs		
Sialy Lewis^x^ mimetic conjugated to PEGylated USPIONs		
Sialy Lewis^x^ mimetic conjugated to USPIONs		
Monoclonal antibody against antigen A7 coupled to ferromagnetic lignosite particles	Antigen A7	
Octreotide conjugated to PEGylated USPIONs	Somatostatin receptors	
Ovarian cancer antigen 183B2 monoclonal antibody conjugated to USPIONs	Ovarian cancer antigen 183B2	
Saposin C-dioleylphosphatidylserine nanovesicles coupled with iron oxides	Phospholipids	
Trastuzumab-dextran iron oxide nanoparticles	HER2 or ErbB2/neu receptor	
Trastuzumab-manganese-doped iron oxide nanoparticles	EGF HER2 receptor	
Z_HER2:342_ Affibody-PEG-SPIONs	EGF	
Single-chain anti-epidermal growth factor receptor antibody fragment conjugated to magnetic iron oxide nanoparticles		
USPIO-Leu-Ile-Lys-Lys-Pro-Phe	Phosphatidylserine	
USPIONs conjugated with Ile-Pro-Leu-Pro-Phe-Tyr-Asn	ß-amyloid (Aß_42_) peptide	
USPIO-anti-CD20 monoclonal antibody	CD20 antigen	
PEG–coated and folic acid–conjugated SPONs	Folate receptor	
*N*-Alkyl-polyethylenimine 2 kDa–stabilized SPIONs	Mesenchymal stem cell labelling	
SPION stabilized by alginate	RES	
Ferumoxsil (Siloxane-coated SPIO)		In vitro, rodents, humans
Citrate-coated (184th variant) SPIONs	Phagocytosis	In vitro, rodents non-primate non-rodent mammals, humans
Ferumoxides (Dextran-coated SPIO)	RES	
Iron oxide nanoparticles-poly-l-lysine complex	-	
Ferumoxtran (USPIONs)	RES	In vitro, rodents, non-primate non-rodent mammals, non-human primates, humans

**Table 2 molecules-26-03437-t002:** Iron oxide-based SPECT contrast agents as described in Molecular Imaging and Contrast Agent Database [15].

Molecule	Source of Signal	Target Region	Level of Research
^99m^Tc-Diethylenetriamine pentaacetic acid SPIONs conjugated with lactobionic acid	Iron oxide, ^99m^Tc	Asialoglycoprotein receptors	In vitro, rodents

**Table 3 molecules-26-03437-t003:** Iron oxide based multimodal contrast agents as described in MICAD [15].

Molecule	Source of Signal	Target	Level of Research
**MRI, SPECT Gamma Planar**			
^111^In-Tetraazacyclododecane-*N*,*N*′,*N*″,*N*‴-tetraacetic acid-benzyl-ChL6-SPIONs	Iron oxide, ^111^In	Antibody-antigen binding	In vitro, rodents
**MRI and PET**			
^124^I-Serum albumin-manganese-magnetism-engineered iron oxide nanoparticles	Iron oxide, ^124^I	Non-targeted	In vitro, rodents
^64^Cu-1,4,7,10-Tetraazacyclododecane-*N*,*N*′,*N*″,*N*‴-tetraacetic acid-iron oxide-c(RGDyK) nanoparticles	^64^Cu and iron oxide	Integrin α_v_β_3_	
**MRI, optical, PET**			
^64^Cu-DTPA-CLIO-VT680	Iron oxides, VT680, ^64^Cu	Macrophages	In vitro, rodents
**MRI, optical NIR fluorescence**			
Annexin V-cross-linked iron oxide-Cy5.5	Iron oxide, Cy5.5	Phosphatidylserine	In vitro, rodents
Anti-vascular cell adhesion molecule monoclonal M/K-2.7 conjugated cross-linked iron oxide-Cy5.5 nanoparticles		Vascular cell adhesion molecule-1	
Cross-linked iron oxide-Cy5.5		Phagocyte and tumour cell	
Lys-Thr-Leu-Leu-Pro-Thr-Pro-cross-linked iron oxide-Cy5.5		Plectin-1	
VCAM-1 internalizing peptide-28 nanoparticles		Vascular cell adhesion molecule-1	
Cys-Arg-Glu-Lys-Ala-SPIO-Cy7 nanoparticles	Iron oxide, Cy7	Clotted plasma proteins	
**Optical, NIR fluorescence imaging and MRI**			
Cy5.5-Amino-terminal fragment of urokinase-type plasminogen activator conjugated to magnetic iron oxide nanoparticles	Iron oxide, Cy5.5	Urokinase-type plasminogen activator receptor	In vitro, rodents
Cy5.5-Arg-Arg-Arg-Arg-crosslinked iron oxide nanoparticle		Proteases	
**MRI, fluorescence molecular tomography (FMT), fluorescence reflectance imaging (FRI)**			
Gly-Ser-Ser-Lys-(FITC)-Gly-Gly-Gly-Cys-Arg-Gly-Asp-Cys-CLIO-Cy5.5	Iron oxides, Cy5.5	α_υ_β_3_ integrin	In vitro, rodents
**MRI, NIFR optical imaging**			
Green fluorescent protein specified small interfering RNA–crosslinked iron oxide nanoparticles-Cy5.5	Iron oxides, Cy5.5	RNAse III	In vitro, rodents
Survivin specified small interfering RNA-CLIO-Cy5.5			
Bombesin peptide conjugated–cross-linked iron oxide-Cy5.5	Iron oxide, Cy5.5, FITC	Gastrin-releasing peptide receptor	
**Optical, near-infrared fluorescence**			
IPLVVPLGGSC (Cy5.5-Cross-linked iron oxide) K(FITC)	Cy5.5, iron oxide	Hepsin	In vitro, rodents
**MRI and optical imaging**			
Rhodamine B isothiocyanate-incorporated, silica-coated magnetic nanoparticle–labelled human cord blood–derived mesenchymal stem cells	SPIO, RITC	Cell imaging	In vitro, rodents

**Table 4 molecules-26-03437-t004:** A summary of biopolymer coating materials for iron oxide cores.

Magnetic Material	Synthesis Method for Iron Oxide Core	Polymeric Coating	Applications	Reference
Fe_3_O_4_	Co-precipitation	Chitosan	MRI	[69,70,71,72]
		Chitosan/Alginate		[65]
		*N,N,N*-trimethyl chitosan	MRI/PET (^68^Ga)	[73]
		Dextran	MRI	[74,75,76,77]
			MRI/PET (^68^Ga)	[59]
			MRI/SPECT (^99m^Tc)	[78]
			MRI/SPECT (^99m^Tc-dipicolylmie (DPA)-alendronate)	[78]
		Carboxymethyl dextran	MRI	[79]
		BSA		[80]
		BSA/HA		[64]
		Folic Acid-BSA		[81]
		Alginate		[82,83]
	Thermal decomposition	Dextran	MRI/PET (^64^Cu)	[84]
		BSA	MRI	[85]
	Hydrothermal	Alginate/PEI		[86]
	Emulsion polymerization	HSA		[87]
	Emulsification	PLGA/Chitosan/Dextran sulfate		[88]
γ-Fe_2_O_3_	Co-precipitation	Dextran	MRI	[89]
		BSA		[90]
			MRI/Fluorescent (FITC)	[91]
		rHSA	MRI	[92]
	Nucleation	Gelatin	MRI/Fluorescence Imaging	[93]
	Co-precipitation/cross-linking	Alginate	MRI	[94]
IONPs	Co-precipitation	Glycol Chitosan/Heparin	MRI	[95]
		Dextran		[96,97,98,99]
			MRI/PET (^89^Zr)	[100]
			MRI/CT	[101]
			Mapping	[102]
		Dextran/Dextran sulfate	MRI	[103]
		Dextran sulfate		[104]
			MRI/PET (^64^Cu)	[105]
		Carboxymethyl-diethylaminoethyl dextran	MRI	[106]
		Gelatin		[66]
	Thermal decomposition	Oleyl-Chitosan	MRI	[107]
		Heparin		[108]
		BSA		[109]
		HSA		[110,111]
	Microemulsion	Gelatin	MRI	[66]
			MRI/Optical	[112]
	Microwave	Dextran	MRI/PET (^68^Ga)	[113]
	Microwave-assisted	Different heparins of distinct anticoagulant/Anti-heparanase	MRI	[114]
		Dextran		[115]
			MRI/PET (^64^Cu)	[116]
	Nucleation	HSA	MRI	[117]
			NIRF	[118]
SPIONs	Co-precipitation	Chitosan	MRI	[119]
		Cationic/Anionic chitosan derivatives		[120]
		Bioactive-conjugated *N*-palmitoyl chitosan		[121]
		Heparin		[122]
		Unfractionated heparin		[123]
		Dextran		[60,124,125,126,127,128,129]
			MPI	[130,131]
		Dextran sulfate	MRI	[132]
		Carboxydextran		[133]
			PET (^89^Zr)/SPECT (^99m^Tc)	[134]
		Carboxymethyl dextran	MRI	[135,136]
			MRI/SPECT (^111^In)	[137]
			PET (^89^Zr)	[138,139]
			PET (^89^Zr)/SPECT (^111^In)	[140]
			PET(^64^Cu)/SPECT (^111^In)	[140]
		Carboxymethyl dextran/Fucoidan	MRI	[141]
		HSA		[142]
		Alginate		[143]
	Thermal decomposition	*N*-palmitoyl-*N*-monomethyl-*N,N*-dimethyl-*N,N,N*-trimethyl-6-O-glycolchitosan	MRI	[144]
		Succinylated heparin monolayer		[145]
		Amphiphilic starlike dextran		[146]
		PLGA/Glycol chitosan	MRI/SPECT (^99m^Tc)	[147]
	Sol gel	Cationic derivative of chitosan/Hyaluronic acid-Curcumin conjugate	MRI	[148]
	Microemulsion	Dextran-b-oligo (amidoamine)		[149]
	Co-precipitation/Cold gelation	Dextran		[60]
	Alkaline co-precipitation in a microfluidic droplet reactor	Dextran	MRI	[150]
USPIOs	Thermal decomposition	Gelatin	MRI	[151]

**Table 5 molecules-26-03437-t005:** Summarization of bio-inspired coating materials for iron oxide cores.

Magnetic Material	Synthesis Method for Iron Oxide Core	Polymeric Coating	Applications	Reference
Fe_3_O_4_	Co-precipitation	PEG	MRI	[178,179]
			SPECT (^111^In)	[180]
		PEG diacid (HOOC–PEG–COOH)	MRI	[181]
		PLA/PEG/D-glucosamine		[182]
		Silica	MRI (Gd-DTPA)	[183]
			MRI	[184]
			MPI	[185]
			MRI/PET (^11^C)	[186]
		Mesoporous Silica	MRI	[187]
		APTES	Tumour imaging agent (^99m^Tc)	[188]
Fe_3_O_4_/γ-Fe_2_O_3_		Poly(2-acrylamido-2-methylpropane sodium sulfonate) P(AMPS)	MRI	[189]
	Thermal decomposition	PEG	MRI	[190,191,192]
			MRI/SPECT (^99m^Tc)	[193]
			MRI/SPECT (^125^I)	[106,194]
			MRI/PET (^68^Ga)	[195]
			MRI/PET (^64^Cu)	[196]
			MRI/PET (^71^As)	[197]
			SPECT (^99m^Tc)	[174]
		PEG-phosphate	MRI	[198]
		Hydroxyl−PEG−Phosphonic Acid		[199]
		DSPE-mPEG_2000_/DSPE-mPEG_2000_ amine	MRI/PET (^68^Ga)	[200]
		Silica	MRI	[201,202]
			MRI/US	[203]
		Mesoporous Silica	MRI	[202,204]
			MRI/Fluorescence Imaging	[190]
		Silica	MRI/Optical (FITC)	[205]
	Hydrothermal	PEG	MRI/PET (^64^Cu)	[206]
		DPPE-mPEG_2000_/DSPE-cPEG_2000_	SPECT (^67^Ga)	[207]
		PEI	MRI	[208]
		Silica		[209]
			MPI	[210]
	Solvothermal	Mesoporous Silica	MRI	[211]
	Sol gel	Silica		[212]
	Microemulsion	PEG		[213]
	Reduction–precipitation	PEI		[214]
	Polyol	PEG bis(carboxymethyl) ether		[215]
		PEG diacid (HOOC-PEG-COOH)	MRI (Gd)	[216]
γ-Fe_2_O_3_	Co-precipitation	PEG-based liposome	MRI/PET (^68^Ga)	[217]
		Silica	MRI	[218]
	Sol gel			[219]
IONPs	Co-precipitation	PEG	MRI	[213]
		PEG-Maleimide	MRI/PET (^64^Cu)	[220]
		PEG/Terminated polystyrene/Linoleic acid	MRI	[221]
		Poly(4-vinylpyridine)/PEG acrylate	MRI/SPECT (^111^In)	[222]
		Silica/Silica-PEG	MRI	[223]
		Silica/PEG	MRI/PET (^68^Ga)	[224]
			MRI/PET-CT (^68^Ga)	[225]
		Poly(glycerol adipate)	MRI	[226]
		PLGA	SPECT (^111^In)	[176]
	Thermal decomposition	PEG	MRI	[227,228]
			MRI/PET (^64^Cu)/ Photoacoustic Tomography (PAT)	[229]
		PEG diacrylate	MRI	[230]
		Amine-terminated PEG (NH_2_-PEG-NH_2_)	MPI	[231]
		DSPE-PEG-2000	MRI	[228]
		Mesoporous Silica/PEG	MRI/NIRF	[232]
		PVP	MRI	[233]
	Hydrothermal	Mesoporous Silica	MRI	[234]
	Solvothermal	PEG	MRI	[235]
	Sol gel	PEI/Silica	MRI	[236]
	Reverse microemulsion	Silica	MRI/NIRF	[139]
			MRI	[237,238]
	Chemical reduction			
SPIOs	Co-precipitation	PEG	MRI	[15,239,240,241]
		PEG/Poly(gallol)		[242]
		PEG/Gallic acid/Dextran	MPI	[130]
		DSPE-mPEG_2000_	SPECT (^111^In)	[243]
		PEI-b-PCL-b-PEG	MRI	[244]
		Silica	Contrast agents in biomedical photoacoustic imaging	[245]
		Mesoporous Silica	MRI/CT/Fluorescence	[246]
	Thermal decomposition	PEG	MRI	[247,248]
			MPI	[249]
		PEG methyl ether	MRI	[250]
		Amine-terminated PEG (NH_2_-PEG)		[251]
		HA/mPEG-succinimidyl succinate		[252]
		Silica		[253]
			MRI/SPECT (^125^I)	[254]
			MRI/PET (^64^Cu)	[255]
		Mesoporous Silica	MRI/Optical	[256]
		Silica/Hyaluronic acid (HA)	MRI	[257]
	Sol gel	Silica		[258]
	Polyol	PEG		[259]
USPIOs	Co-precipitation	Silica/APTMS/AEAPTMS	MRI	[260]
	Thermal decomposition	PEG	MRI/ SPECT(^99m^Tc)	[261]
	Microwave	Phosphonate-PEG (PO-PEG-NH_2_)	MRI	[262]
	Polyol	PEG	MRI/Optical (FITC)	[263]

**Table 6 molecules-26-03437-t006:** Examples of animal studies assessing the use of iron oxide-based nanoparticles in imaging.

Core-Shell Iron Oxide Agent	Imaging Modality	In Vivo Model	Reference
SPIONs coated with silica, APTES and (3-glycidoxypropyl) methyldiethoxysilane	MRI: medium T_2_ along with a minor effect on T_1_	Zebrafish (Danio rerio) model	[271]
Fe_3_O_4_ nanoparticles stabilized by PVP, trisodium citrate, and maleic anhydride	MRI-T_2_	White rabbit	[272]
IONPs coated with PEG		Tumour xenograft model on female NOD-SCID IL2 mice with human breast cell line (MDA-MB-231)	[235]
Luteinizing hormone-releasing hormone-conjugated PEG-coated magnetite nanoparticles		BALB/c nude mice (Triple negative breast cancer model)	[273]
Magnetic fibrin nanoparticles conjugated with folic acid		Albino Wistar rats with antigen-induced arthritis	[274]
HSA-coated iron oxide nanoparticles		BALB/c mice (breast cancer model with MDA-MB-231 cells)	[142]
PLA-PEG-D-glucosamine Fe_3_O_4_ nanoparticles	MRI-T_2_*	Tumour-bearing BABL/c mice inoculated with 4 T_1_ cells	[182]
PEG/PEI-SPIONs conjugated with transferrin		Kunming mice brain	[275]
USPIOs coated with carboxymethyldextran and coupled or not with a low-molecular-weight aminated fucoidan		Elastase-induced Wistar rat model	[141]
SPIONs coated with dextran and functionalized with anti-insulin-like-growth-factor binding protein 7 (anti-IGFBP7)	MRI-T_2_ and T_2_*	Nude CD-1 mouse model of glioblastoma multiforme	[276]

**Table 7 molecules-26-03437-t007:** Iron oxide-based nanoparticles currently under evaluation for human use in imaging (up to 10 March 2021) [277].

Material	Disease	Imaging Technique	Status
IONPs	Lymph node metastases	MRI	Phase I, Withdrawn (NCT02689401)
	Healthy subjects-to determine MTD, Pharmacokinetic, safety/tolerability		Phase I, Completed (NCT02744248)
SPIONs	Lymph node metastases		Phase IV, Completed (NCT00920023)
	Sentinel lymph nodes		Completed (NCT03243435)
			Phase IV, Unknown (NCT02612870)
			Phase III, Recruiting (NCT04722692)
			Unknown (NCT03449615)
	Melanoma		Not Applicable, Completed (NCT03898687)
	Healthy volunteer-cell tracking		Not Applicable, Unknown (NCT00972946)
	Primary and metastatic hepatic malignancies	MRI/SPECT (^99m^Tc)	Recruiting (NCT04682847)
USPIONs	Lymph nodes	MRI	Early Phase 1, Recruiting (NCT02857218)
	Lymph node metastases		Phase III, Recruiting (NCT04261777)
			Not Applicable, Completed (NCT01815333)
			Not Applicable, Recruiting (NCT04311047)
	Malignant pelvic lymph nodes		Not Applicable, Terminated (NCT00147238)
	Pelvic lymph nodes		Early Phase 1, Recruiting (NCT03280277)
	Pelvic nodal metastases		Phase I and II, Completed (NCT00188695)
	Papillary carcinoma of thyroid gland/metastatic medullary thyroid cancer/follicular thyroid cancer lymph node metastasis		Not Applicable, Completed (NCT01927887)
	Brain neoplasms		Phase II, Terminated (NCT00659334)
	Oesophageal neoplasms		Not Applicable, Completed (NCT02253602)
	Squamous cell carcinomas		Phase 1, Active (NCT01895829)
	Ischemic heart disease		Phase I, Completed (NCT03651791)
	Multiple sclerosis		Phase I, Completed (NCT02511028)
	Aortic dissection		Recruiting (NCT03948555)
	Myocardial infarction		Phase II, Unknown (NCT01995799)
			Not Applicable, Completed (NCT01323296)
	Myocardial cellular inflammation		Phase II and III, Completed (NCT02319278)
	Abdominal aortic aneurysm		Terminated (NCT01749280)
	Atherosclerosis		Unknown (NCT01674257)
	Coronary artery disease	MRI/PET	Recruiting (NCT03451448)

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
