# Peer review of "Imaging Constructs: The Rise of Iron Oxide Nanoparticles"

_molecules, 2021, doi:10.3390/molecules26113437_

Round 1
Reviewer 1 Report
The purpose of this review is to give an overview of the potential of magnetic iron oxides in diagnostics imaging. The manuscript focuses on MRI alone and in multimodal combination with radio-imaging and optical imaging. The subject is certainly important, and the authors have gathered a significant amount of papers on this subject. It is particularly interesting the update on the formulations that have reached clinical trials.
However, the information is mostly offered as a list of publications with a brief description of their contents. The manuscript lacks a comprehensive discussion including a historical perspective or a classification of the different materials accordingly to the particle size, type of coating or clinical performance, which would be very helpful for the reader. Accordingly with the aims of the manuscript, a deeper analysis of the information should be expected. For instance, the literature revision could be accompanied by a comparative evaluation of the efficacy of the different reported materials (in terms of blood toxicity, biodistribution, targetting efficacy, imaging contrast, relaxivities, etc.) in relation with their structural features, some guidelines for the enhancement of their performance, and the future perspectives of this area of research.
Moreover, there are some points that could be improved. For instance, the descriptions of the iron oxide nanoparticle synthesis, coating procedures and multifunctionalization are not very precise, and there are some important approaches that are missing such as the polyols, electrochemical, aerosols, magnetotactic bacteria, etc.
The English language should also be improved, especially the syntax, which is sometimes misleading.
Author Response
Response to Reviewer 1 Comments
On behalf of the manuscript authors, I would like to sincerely thank you for your kind answer and valuable advices given to improve this review paper. All your comments are fare and justified considering the body of the text. All the modifications were made with green color. The answers to the reviewer comments are listed below, as follows:
Comments and Suggestions of Reviewer 1 for Authors
The purpose of this review is to give an overview of the potential of magnetic iron oxides in diagnostics imaging. The manuscript focuses on MRI alone and in multimodal combination with radio-imaging and optical imaging. The subject is certainly important, and the authors have gathered a significant amount of papers on this subject. It is particularly interesting the update on the formulations that have reached clinical trials.
However, the information is mostly offered as a list of publications with a brief description of their contents.
Point 1:
The manuscript lacks a comprehensive discussion including a historical perspective or a classification of the different materials accordingly to the particle size, type of coating or clinical performance, which would be very helpful for the reader.
Response 1: The review paper presents the classification of the potential/used contrast agents based on the iron oxide type, namely, SPIONs, USPIONs, IONPs, Fe2O3, and so on. There are different review papers published in the last year that deal with the classification of iron oxide materials in general based on the type of coating, type of system, size, applications potential as listed and described in Section 3. We added a Figure to describe all the identified possibilities for coating of iron oxide nanoparticles used as diagnostic agents.
Point 2:
Accordingly with the aims of the manuscript, a deeper analysis of the information should be expected. For instance, the literature revision could be accompanied by a comparative evaluation of the efficacy of the different reported materials (in terms of blood toxicity, biodistribution, targeting efficacy, imaging contrast, relaxivities, etc.) in relation with their structural features, some guidelines for the enhancement of their performance, and the future perspectives of this area of research.
Response 2: As recommended the authors added relevant information and also perspective potential as envisioned by us.
Point 3:
Moreover, there are some points that could be improved. For instance, the descriptions of the iron oxide nanoparticle synthesis, coating procedures and multifunctionalization are not very precise, and there are some important approaches that are missing such as the polyols, electrochemical, aerosols, magnetotactic bacteria, etc.
Response 3:
In this review paper we intend to describe only the most reported processes for the fabrication of iron oxide nanoparticles as diagnostic platforms, such as co-precipitation, thermal decomposition, hydrothermal, sol-gel and microemulsion methods. Indeed, some important approaches were missing; therefore, we added polyol, magnetotactic bacteria, aerosols, electrochemical and microfluidic methods in the manuscript, as recommended, since we found reports of iron oxides obtained through these methods as diagnostic vehicles. The reference list was updated accordingly.
Point 4:
The English language should also be improved, especially the syntax, which is sometimes misleading
Response 4: A native British English revised the entire manuscript in terms of English language.
Reviewer 2 Report
Dear authors,
your manuscript is very well written and covers a vast area of iron oxide based nanoparticles made for imaging. Nonetheless, I have a few minor and major comments:
In general, you show mainly the positive aspects of the candidates but hardly show any downsides of the particles. But to get the whole story both sides have to be addressed.
l. 92, Table 1: Please, fuse some entries that seem to appear twice, for a better readability
l.119f: Please give some examples for physical synthesis strategies, for which the given downsides apply
l.124f: Please give some examples for biological procedures. Especially for biological techniques scalability is not always as easy as described. Also the characteristic of “low cost” and “laborious and time-consuming” don’t fit together.
l.132f: What do you mean with “eco-friendly”? In terms of temperature and energy cost I may agree, but often harsh chemicals have to be used for the synthesis (Examples: NH3, Epichlorohydrin, etc.). In addition co-precipitation does not always result in “black” iron oxides. γ-Fe2O3 is not black either.
l.138f: Khalafalla et al mentioned it earlier: Khalafalla, S.E. and G.W. Reimers, Preparation of dilution-stable aqueous magnetic fluids. Ieee Transactions on Magnetics, 1980. 16(2): p. 178-183.
l.152: What is a “positive toxicity profile”?
l.235, Table4: The differentiation between IONPs, SPIOs etc. is not always clear. Also synthesis method is also not always clear, e.g. crosslinking and cold gelation also involve a co-precipitation step.
L268: please use SI units, even if you cite this value! I know that many magnetic related measurements are not given in SI units, but this is a practice that should be avoided.
L302f: Dextran has α-D 1,6 linkages in the back bone and α-D 1,4 or sometimes α-D 1,3 linkages at the branching points
L547ff, section 3.2: I really don´t understand what the examples given here has to do with “bio-inspired” polymers or “mimic[king] the biological systems”. What makes a silica or PEG-coated Particles more “bio-inspired” than a Dextran or HSA coated particle?
l.680ff: PEG is definitely not as “non-immunogenic” as it is stated here, especially when its bound on the surface of a nanoformulation. See for example:
Mohamed M, Abu Lila AS, Shimizu T, Alaaeldin E, Hussein A, Sarhan HA, Szebeni J, Ishida T. PEGylated liposomes: immunological responses. Sci Technol Adv Mater. 2019 Jun 26;20(1):710-724. doi: 10.1080/14686996.2019.1627174. PMID: 31275462; PMCID: PMC6598536.
Kozma GT, Shimizu T, Ishida T, Szebeni J. Anti-PEG antibodies: Properties, formation, testing and role in adverse immune reactions to PEGylated nano-biopharmaceuticals. Adv Drug Deliv Rev. 2020;154-155:163-175. doi: 10.1016/j.addr.2020.07.024. Epub 2020 Aug 1. PMID: 32745496.
Dézsi L, Fülöp T, Mészáros T, Szénási G, Urbanics R, Vázsonyi C, Őrfi E, Rosivall L, Nemes R, Kok RJ, Metselaar JM, Storm G, Szebeni J. Features of complement activation-related pseudoallergy to liposomes with different surface charge and PEGylation: comparison of the porcine and rat responses. J Control Release. 2014 Dec 10;195:2-10. doi: 10.1016/j.jconrel.2014.08.009. Epub 2014 Aug 19. PMID: 25148822.
l.760, Table 7: The differentiation between IONPs, SPIOs etc. is not always clear. Also, in order to interpret the list it would be valuable to add the date of the last update of the different studies
l.762f: Please also mention the lack of the “safe-by-design” mindset of many groups.
l.767ff: The review should also cover the critical facts about ferumoxytol, including fatal outcomes upon administration, and also FDA warnings etc.
l.792ff, chapter 6: I highly recommend to merge your work into the rest of the review to the respective parts were they belong to.
l.807: SPION formulation tend to interfere with the MTT assay and should be avoided with these substances.
Author Response
Response to Reviewer 2 Comments
On behalf of the manuscript authors, I would like to sincerely thank you for your kind answer and valuable advices given to improve this review paper. All your comments are fare and justified considering the body of the text. All the modifications were made in green font color. The answers to the reviewer comments are listed below, as follows:
Comments and Suggestions of Reviewer 2 for Authors
Dear authors,
Your manuscript is very well written and covers a vast area of iron oxide based nanoparticles made for imaging. Nonetheless, I have a few minor and major comments:
Point 1:
In general, you show mainly the positive aspects of the candidates but hardly show any downsides of the particles. But to get the whole story both sides have to be addressed.
Response 1:
Point 2:
- 92, Table 1: Please, fuse some entries that seem to appear twice, for a better readability
Response 2: Table 1 was revised and all the identical entries were fused.
Point 3:
l.119f: Please give some examples for physical synthesis strategies, for which the given downsides apply
Response 3: The physical methods for iron oxide synthesis were added.
Point 4:
l.124f: Please give some examples for biological procedures. Especially for biological techniques scalability is not always as easy as described. Also the characteristic of “low cost” and “laborious and time-consuming” don’t fit together.
Response 4: The microbial incubation method was added as the biological approach that has the mentioned advantages and drawbacks, also as mentioned in the reference at the begging of Section 2.
Point 5: l.132f: What do you mean with “eco-friendly”? In terms of temperature and energy cost I may agree, but often harsh chemicals have to be used for the synthesis (Examples: NH3, Epichlorohydrin, etc.). In addition co-precipitation does not always result in “black” iron oxides. γ-Fe2O3 is not black either.
Response 5: The paragraph was modified according to the reviewer’s comment. I agree with the term partial eco-friendly method, and indeed is not always black, the one that I obtain is black.
Point 6: l.138f: Khalafalla et al mentioned it earlier: Khalafalla, S.E. and G.W. Reimers, Preparation of dilution-stable aqueous magnetic fluids. Ieee Transactions on Magnetics, 1980. 16(2): p. 178-183.
Response 6: Thank you for this information. The reference was changed accordingly.
Point 7: l.152: What is a “positive toxicity profile”?
Response 7: It refers to the non-toxic profile of the obtained nanoparticles.
Point 8: l.235, Table4: The differentiation between IONPs, SPIOs etc. is not always clear. Also synthesis method is also not always clear, e.g. crosslinking and cold gelation also involve a co-precipitation step.
Response 8: The Table was revised along with the synthesis methods for magnetic core and useful information was added.
Point 9: L268: please use SI units, even if you cite this value! I know that many magnetic related measurements are not given in SI units, but this is a practice that should be avoided.
Response 9: The SI unit for emu/g is Am2/kg, and was changed as you suggested.
Point 10: L302f: Dextran has α-D 1,6 linkages in the back bone and α-D 1,4 or sometimes α-D 1,3 linkages at the branching points
Response 10: Corrected.
Point 11: L547ff, section 3.2: I really don´t understand what the examples given here has to do with “bio-inspired” polymers or “mimic[king] the biological systems”. What makes a silica or PEG-coated Particles more “bio-inspired” than a Dextran or HSA coated particle?
Response 11: The term was replaced with synthetic biocompatible polymers, along with the whole paragraph modifications.
Point 12: l.680ff: PEG is definitely not as “non-immunogenic” as it is stated here, especially when its bound on the surface of a nanoformulation. See for example:
Mohamed M, Abu Lila AS, Shimizu T, Alaaeldin E, Hussein A, Sarhan HA, Szebeni J, Ishida T. PEGylated liposomes: immunological responses. Sci Technol Adv Mater. 2019 Jun 26;20(1):710-724. doi: 10.1080/14686996.2019.1627174. PMID: 31275462; PMCID: PMC6598536.
Kozma GT, Shimizu T, Ishida T, Szebeni J. Anti-PEG antibodies: Properties, formation, testing and role in adverse immune reactions to PEGylated nano-biopharmaceuticals. Adv Drug Deliv Rev. 2020;154-155:163-175. doi: 10.1016/j.addr.2020.07.024. Epub 2020 Aug 1. PMID: 32745496.
Dézsi L, Fülöp T, Mészáros T, Szénási G, Urbanics R, Vázsonyi C, Őrfi E, Rosivall L, Nemes R, Kok RJ, Metselaar JM, Storm G, Szebeni J. Features of complement activation-related pseudoallergy to liposomes with different surface charge and PEGylation: comparison of the porcine and rat responses. J Control Release. 2014 Dec 10;195:2-10. doi: 10.1016/j.jconrel.2014.08.009. Epub 2014 Aug 19. PMID: 25148822.
Response 12: The paragraph was reformulated and the reference list updated.
Point 13: l.760, Table 7: The differentiation between IONPs, SPIOs etc. is not always clear. Also, in order to interpret the list it would be valuable to add the date of the last update of the different studies
Response 13: The Table was revised and the data was added.
Point 14: l.762f: Please also mention the lack of the “safe-by-design” mindset of many groups.
Response 14: The term was added.
Point 15: l.767ff: The review should also cover the critical facts about ferumoxytol, including fatal outcomes upon administration, and also FDA warnings etc.
Response 15: Critical facts about ferumoxytol were added.
Point 16: l.807: SPION formulation tend to interfere with the MTT assay and should be avoided with these substances.
Response 17: In vitro cell viability assay displayed good biocompatibility of the conjugates with the percentage of viable cells exceeding 80% after 48 h incubation for normal cell line in a medium that contains bovine serum albumin. The information was modified.
Round 2
Reviewer 1 Report
The authors made an effort to improve the manuscript and I think it is now suitable for publication.